# Exploring the Impact of Rapidly Actuated Control Surfaces on Drone Aerodynamics

Ashim Panta * , Matthew Marino †, Alex Fisher †, Abdulghani Mohamed † and Simon Watkins †

RMIT UAS Research Laboratory, RMIT University, Bundoora 3083, Australia;
matthew.marino@rmit.edu.au (M.M.); alex.fisher@rmit.edu.au (A.F.); abdulghani.mohamed@rmit.edu.au (A.M.);
simon.watkins@rmit.edu.au (S.W.)
* Correspondence: ashim.panta14@gmail.com
† These authors contributed equally to this work.

**Abstract:** This study investigates the use of rapidly actuated leading-edge and trailing-edge control surfaces to improve the control authority of small fixed-wing drones. Static and dynamic characteristics were investigated and presented in two separate papers. In this paper, the focus is on the dynamic effects observed from rapidly actuated 30% chord leading- or trailing-edge hinged control surfaces affixed to two flat-plate airfoils. Forces were resolved from surface pressure measurements and are augmented by PIV measurements, smoke flow visualization and analyses. The static study revealed that trailing-edge control surfaces exhibited higher effectiveness in producing time-averaged $C_L$ compared to leading-edge control surfaces. However, leading-edge control surfaces exhibit significantly less fluctuation in pressure and lift coefficients at fixed angles of attack and control surface deflections, indicating better stability. Unsteady aerodynamic effects of the airfoil at $\alpha = 0°$ and "ramp" deflections of trailing- and leading-edge control surfaces from 0° to 40° with variations in actuation rates showed that $C_L$ peaks are approximately three to four times greater than static values for the case of the leading-edge control surface. This has significant implications for fixed-wing drone maneuverability and countering the effects of atmospheric turbulence.

**Keywords:** leading-edge control; control surface; rapid actuation

## 1. Introduction

Small fixed-wing drones are significantly challenged by disturbances from turbulence in the atmosphere or from the rapid changes in the angle of attack experienced when flying through the shear layers associated with flows around obstacles [1–4]. Existing turbulence methods were found to be limited by the response rate and authority of control actuators [5–12]; however, these studies investigated only with conventional trailing-edge control surfaces as opposed to leading-edge control surfaces (hereafter called TECS and LECS).

Existing static studies of various leading-edge devices revealed that leading-edge flaps could improve an airfoil's performance at low Reynolds number [13–18]. The static lift characteristics of LECS were extensively studied across angles of attack from 0° to 30° and control surface deflection ranging from 0° to ± 45° [19]. A reversal in lift was noted beyond specific control surface deflections (e.g., at $\alpha = 0°$, $C_L$ reversed beyond $\psi = \pm20°$). Though the time-averaged maximum $C_L$ for any given control surface deflection was significantly greater with TECS deflections than LECS, LECS was found to be advantageous with regards to the time-varying lift characteristics [19]. Thus, the use of leading-edge control surfaces has emerged as a promising area of study for improving airfoil performance on fixed-wing drones. However, the dynamic characteristics of LECS in low-Reynolds-number flight are not well understood in the current body of literature.

Consider moments taken about a pivot point when deflecting TECS or LECS; there are aerodynamic loads (mainly arising from the surface pressures) and inertial loads (from the acceleration of the control surface mass). In the case of conventional TECS, the aerodynamic forces generate a "stable" or restoring load, contrary to LECS. Due to the unstable nature of LECS and with TECS providing the required control response for large aircraft, LECSs have not been widely used to date. Since small fixed-wing drones experience relatively smaller loads and require control inputs at higher frequencies, LECS could potentially be useful to achieve rapid control response and authority needed for turbulence rejection. A qualitative study of LECS as a primary control surface revealed that rapid deflections (non-dimensional actuation rate, $\dot{\psi} \geq 0.1$) promoted flow attachment on the airfoil [20,21]. In these studies, higher deflection rates were found to develop Leading-Edge Vortices (LEVs), which grew in size as they traversed across the chord. The presence of these LEVs, which have a region of low pressure in the vortex core on the upper surface of the airfoil, is known to provide significant transient lift increment. The size and strength of the vortex core and its traversing rate across the airfoil chord were found to correlate with the actuation rate [21]. Since these LEV characteristics influence the amount of lift produced, it is expected that lift production can be augmented by manipulating the kinematics of LECS. Thus, this novel control technique could potentially allow small fixed-wing drones to produce sufficient control responsiveness and authority to mitigate the high-frequency perturbations from turbulence.

Existing studies by Mancini [22] and Rennie [23] have investigated the flow mechanics associated with a dynamically actuated TECS. Mancini used a 50% TECS on a NACA airfoil (aspect ratio of 4) and conducted PIV analysis and direct force measurements to provide insight into the unsteady aerodynamics associated. Rennie conducted smoke flow visualization and took surface pressure measurements on a similar NACA airfoil (two-dimensional) with a 27% TECS. From these two studies, it was found that the $C_L$ response begins immediately with the start of the TECS deflection. Even when deflecting to large angles, the flow conforms to the changing geometry and remains attached. Thus, $C_L$ increases in a nominally linear fashion with control surface deflection until it peaks at a maximum value (typically one to two times the steady-state value) and then eventually relaxes to its steady-state conditions. Irrespective of the initial flow condition (attached or separated), a relatively large transient $C_L$ was observed during the control surface deflection history, even at large deflection angles [22,23]. Whether or not the same can be said for LECS remains unanswered by the existing literature. Hence, the objective of this paper is to investigate the flow mechanics and dynamic forces generated by using LECS and TECS at a range of deflection rates at $Re = 40,000$ (chord-based). In the latter part of the paper, an existing unsteady potential flow model by Theodorsen is reviewed and modified to represent the dynamic lift responses arising from rapid actuation of LECS and TECS.

## 2. Experimental Setup

### 2.1. Wind Tunnels and Apparatus

Two flat-plate airfoils with elliptical leading edges were used. One featured an LECS and the other a TECS as depicted in Figure 1 (right). The airfoils were manufactured using stereolithography, and 1 mm PVC tubes with a wall thickness of 0.4 mm were inserted into the airfoil. The control surfaces were hinged to the wing body using a pin joint. The gap on the hinge line was approximately 0.5 mm and was sealed with flexible rubber tape. Due to practical limitations, the total number of surface pressure taps was limited to 37, of which 18 were on the top surface, 18 on the bottom surface, and one at the leading edge. Tap diameters were 1 mm (driven by printing resolution limitations) and were located from the leading edge to as close to the sharp trailing edge as possible; see Figure 1 (right).

Smoke flow visualizations and pressure measurements were gathered in the RMIT Aerospace Wind Tunnel (RAWT), and particle image velocimetry (PIV) experiments were conducted at the Hao Liu Laboratory Wind Tunnel (HLWT) at Chiba University.

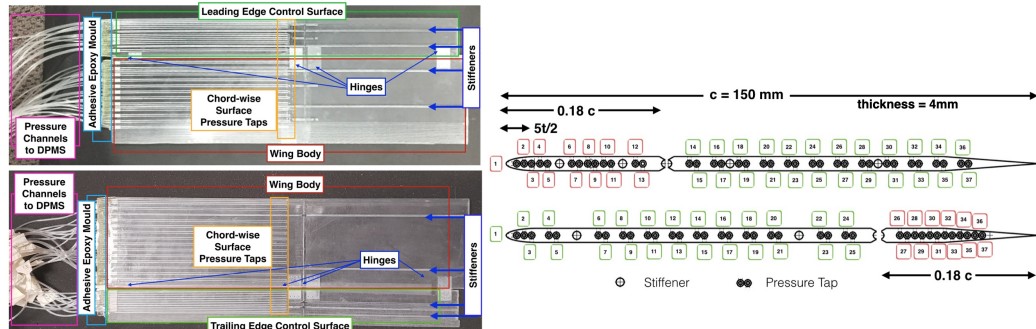

**Figure 1.** Schematic top view (**left**) and cross-sectional view (**right**) of the two pressure-tapped flat-plate airfoil set-ups (labels numbered 1–37 show the locations of the pressure taps).

RAWT is a closed-return tunnel and has a octagonal test section of 2.1 m, 1.3 m, and 1.1 m [L × W × H]. The 4:1 contraction section of the RAWT, before the test section, is equipped with a flow straightening honeycomb screen. The operating velocity range of the test section of the RAWT was from 1 to 45 m/s. Turbulence intensity of the empty cross-section was measured to be ≤1.2% by [24,25] for flow velocity ranging from 0 to 12 m/s. A top speed of 50 m/s could be achieved within the test section; however, for the current low-*Re* experiments, the maximum testing velocity was only 12 m/s. The tunnel was equipped with a differential pressure measuring system, onto which a pitot-static tube was connected near the inlet of the test section to monitor the flow velocities. An additional pitot tube was also mounted on the rig alongside the leading edge of the test airfoil to record the local dynamic pressure; see Figure 2.

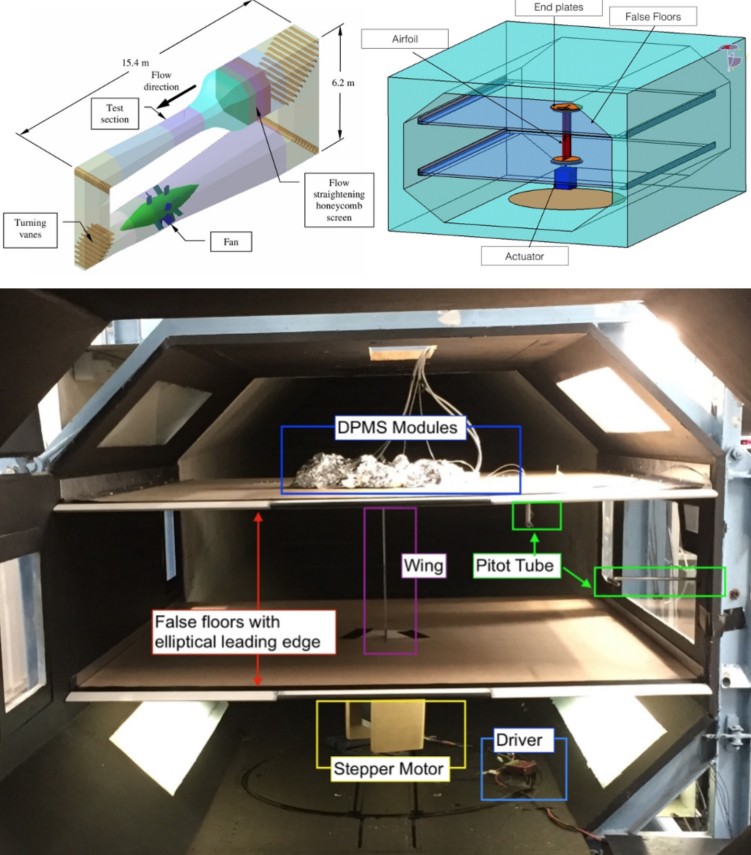

**Figure 2.** Perspective view of RMIT Aerospace Wind Tunnel (**top-left**) and test set-up (**top-right**). Schematic front view of the test section of the wind tunnel set-up with test apparatus (**bottom**).

An insert box was designed to provide a nominally two-dimensional flow case. This consisted of two false floors of dimensions 1.5 m, 1.8 m, and 0.98 m [L × H × W]. Elliptical leading edges were added to the false floor to prevent flow separation. The false floors had 170 mm holes cut out from their center to allow the wing mounting. The airfoil and control surfaces were inserted into two round end-plates which were then inserted into the false floors to permit angles of attack to be varied.

The HLWT at Chiba University has a test section of 1 m, 1 m, and 2 m [L × W × H] and is capable of achieving a smooth airflow (≤2% turbulence) ranging from 0.5 m/s to 11.0 m/s [26]; see Figure 3. Flow velocities were monitored using a hot-wire anemometer inserted into the cross-section of the tunnel. A similar insert box to that used in the RAWT was manufactured to provide nominally two-dimensional flow. The insert box, shown in Figure 4 (right), had dimensions 1.2 m, 0.42 m, and 0.98 m [L × W × H].

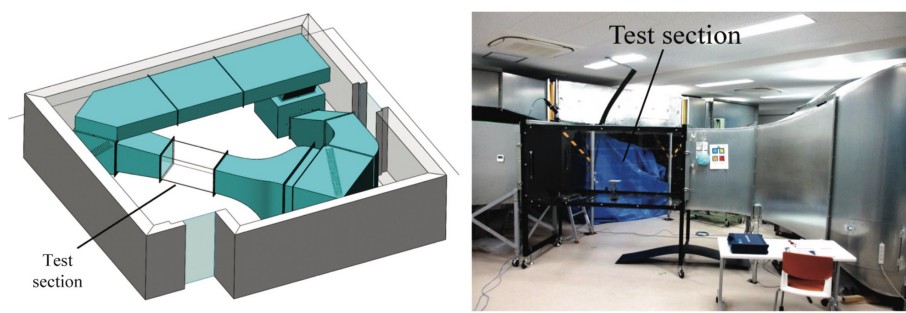

**Figure 3.** Perspective view of Hao Liu laboratory wind tunnel [26] (**left**) and a close-up photo of the test section (**right**).

Near-field flow structures around the airfoil were measured with a PIV system equipped into the test section; see Figure 4. The airfoil was located horizontally at the center of the test section with its angle of attack controlled by the turn-plate. The suction surface of the airfoil was illuminated by a laser light generated by a Nd-YAG pulsed laser system (LDP-100 MQG, 532 nm, Lee Laser, Inc., USA), which was guided to the top of the test section via an optic fiber and then diverged into a two-millimeter-wide band sheet via a cylindrical lens. As a tracer particle, DEHS (di-2-Ethylhexyl sebacate) mist was sprayed in the wind tunnel at a diameter of approximately 1 μm by a particle generator (PivPart14, PivTech GmbH, Germany). The laser sheet was produced with a pulse separation interval of 120 μs for a wind speed of 4 m/s. To ensure that the testing conditions remained similar at both wind tunnels, a chord-based Reynolds number was maintained from tunnel velocity and atmospheric conditions. When the blockage ratio is greater than 3–5%, appropriate blockage correction needs to be applied. In this case, the cross-sectional area of the RAWT and HLWT are 1.43 m$^2$ and 2 m$^2$, respectively, while the maximum frontal area of the models is 0.031 m$^2$ and 0.035 m$^2$, which gives a solid blockage ratio of ≤2% (even at maximum airfoil and control surface deflection). Thus, no blockage correction was deemed necessary.

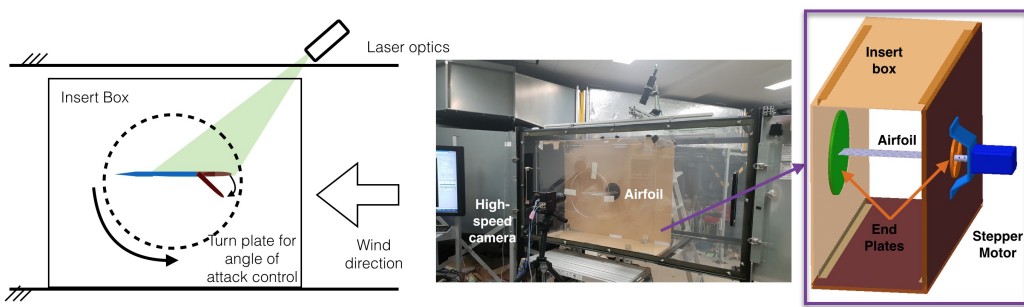

**Figure 4.** Schematic of the PIV setup (**left**), photo (**center**), and insert box (**right**).

### 2.2. Control Surface Deflection Kinematics

A stepper motor was utilized to adjust the control surface deflections at increments of $0.18°$. The accuracy of the stepper motor was specified by the manufacturer as $\leq\pm0.1°$ non-cumulative. The wing's angle of attack was adjusted manually and referenced from $0°$ (set by balancing the upper and lower surface pressure measurements). A potentiometer was attached to the hinge line on the side opposite the stepper motor to precisely measure control surface deflections. The accuracy of the potentiometer was specified to be within $\pm0.9°$. A custom control system was developed and used to drive the control surfaces at angular rates ranging from 0.38 to 38 rad/s (non-dimensional actuation rates $\dot{\beta}$, $\dot{\psi}$ between 0.1 and 1.4). Figure 5 illustrates the remarkable consistency observed between the inputs from the stepper motor and the outputs of the potentiometer.

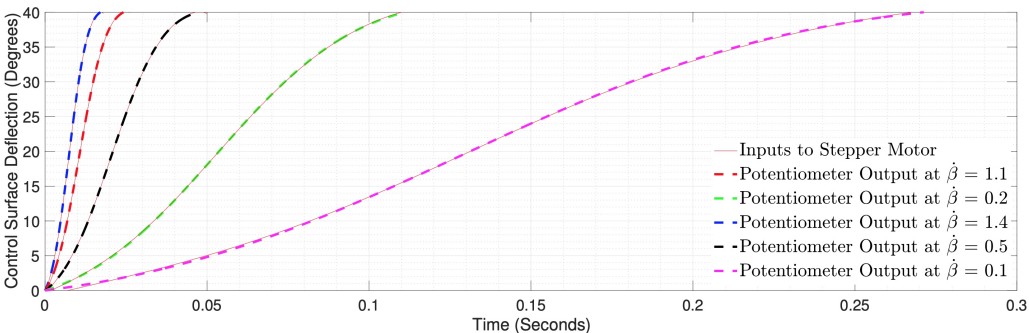

**Figure 5.** Stepper motor input and potentiometer outputs at various actuation rates.

The range of motion histories studied is shown in Figure 5, where $\beta(t) = At^3 + Bt^2$ and $\psi(t) = At^3 + Bt^2$. A and B are constants dependent on reduced frequency ($k$) and deflection angles ($\psi$ or $\beta$). For the purpose of analytical prediction of $C_L$ (presented in Section 3.5), these motion histories were imported into Matlab, where a Discrete Fourier Transform was applied to convert the motion history of TECS or LECS from the time domain into frequency, such that the $C_L$ could be calculated as a function of $k$.

### 2.3. Pressure, Lift, and PIV Flow Field Calculations

To measure surface pressure, a Dynamic Pressure Measurement System (DPMS) was used. The DPMS consisted of 4 modules, each equipped with 15 pressure transducers. Plastic tubing connected the short hypodermic tubes in each channel to the airfoils. With a resolution of 0.001 Pa, the DPMS exhibited a frequency response of several kHz, which depended on the tubing diameter, length, and associated dynamic response. Prior to each run, the transducers were zeroed to eliminate any small drift errors caused by temperature changes. To minimize drift errors, the sensors were warmed up for 1.5 h before experimentation. The length of tubing was carefully chosen to introduce significant distortion to pressure fluctuations propagating through the tubes. To account for the tubing's effect on dynamic pressure measurements, the DPMS software incorporated a calibration method similar to the one described in [25], based on the original work by [27]. Further information on the dynamic calibration process can be found in Appendix B.

$C_P$ was calculated using $C_{Ptop} = \frac{p_{top}-p}{1/2\rho U^2}$ and $C_{Pbottom} = \frac{p_{bottom}-p}{1/2\rho U^2}$ and was integrated using the trapezoidal method to estimate sectional forces using $C_N = \int_0^c (C_{Pbottom} - C_{Ptop})dx$ and $C_L = C_N cos(\alpha)$ (ignoring shear drag effects).

Vector flow field, vorticity, and flow velocity magnitude calculations were performed within the *PIV Lab* application. Further details of the computation mechanism can be found in [28].

### 2.4. Accuracy and Errors

Evaluation of errors in the experimental data is given in the Appendix. The transient lift force calculated from the integrated pressure coefficients was found to vary, and the

magnitude of the variation depended upon actuation speeds. To minimize the effects, each trial was repeated 15 times, and the average was taken. These averaged results are presented in Section 3. The largest variation was at the highest actuation speed, and this is discussed in Appendix D . Due to the high number of repeats, any bias uncertainties are averaged out and impose a negligible influence on the results. Precision errors were also found to be within $\pm 5\%$ of transient forces and moments. It was concluded that the errors and uncertainties were relatively small and thus give confidence in the results presented in the following section.

## 3. Results and Discussion

The purpose of this section is to provide a foundational understanding as to how the flow behaves under dynamic TECS and LECS kinematics in order to provide the context for the detailed measurements and analysis of other test cases in the subsequent sections. An intermediate deflection rate was chosen here as a representative case to investigate the underlying flow mechanics responsible for the transient $C_L$ response.

### 3.1. Surface Pressure and Flow Behavior during a Dynamic Trailing-Edge Control Surface Deflection

Figure 6 presents the $C_L$ response of a dynamically deflected TECS from $0°$ to $40°$ at an intermediate deflection rate of $\dot{\beta} = 0.27$. Upon initiation of the control surface deflection (a), the $C_L$ responds immediately, reaching a maximum value of 1.4 at approximately 80% of the control surface deflection (b). This indicates that the flow adheres closely to the newly acquired camber line, even at significant deflection angles, throughout the control surface transient. Once the deflection ends, the flow separates, and the transient $C_L$ gradually relaxes to a steady-state value, as depicted in Figure 6 (d–f). These findings align with PIV and smoke flow visualization results from previous studies by Mancini and Rennie [22,23], where dynamically actuated TECS exhibited the formation of a large Trailing-Edge Vortex (TEV) upon reaching the end of the deflection, followed by the shedding of several smaller TEVs.

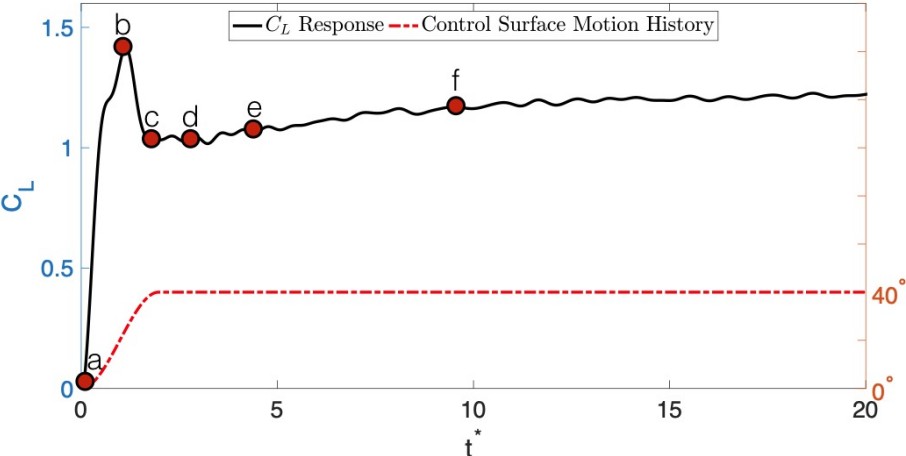

**Figure 6.** Total $C_L$ (i.e., airfoil and control surface) as a function of convective times. Airfoil angle of attack = $0°$, TECS deflected from $0°$ to $40°$ at a non-dimensional actuation rate, $\dot{\beta} = 0.27$.

Figure 7 shows the chord-wise pressure distribution at various stages of the control surface deflection measured using the techniques described in Section 2.3. Before the motion starts, there is a typical $C_P$ distribution where the top and bottom surface pressures are nearly identical, and the stagnation pressure is at unity; see Figure 7a. As the deflection begins (see Figure 7b), the $C_P$ on the top surface decreases and increases on the bottom surface with the development of a favorable static pressure gradient on the top surface during the control surface deflection. Thus, the flow remains momentarily attached despite

the high deflection angles; see Figure 7c. This was also seen in the PIV and smoke flow visualization experiments by Mancini and Rennie [23,29]. The largest pressure difference between the top and bottom surface is created just past halfway of the deflection history. This correlates well with the largest $C_L$ peak noticed in Figure 6. Figure 7b,c also show that the pressure decreases ($-C_P$) on the upper surface, approaching the hinge location, where the attached flow experiences a sharp turn down the control surface (acceleration due to control surface geometry). Following the end of the control surface motion, the $C_P$ distribution on the top surface gradually flattens just aft of the leading edge; see Figure 7d–f. This suggests that the attached flow gradually separates aft of the hinge and relaxes to a steady-state condition.

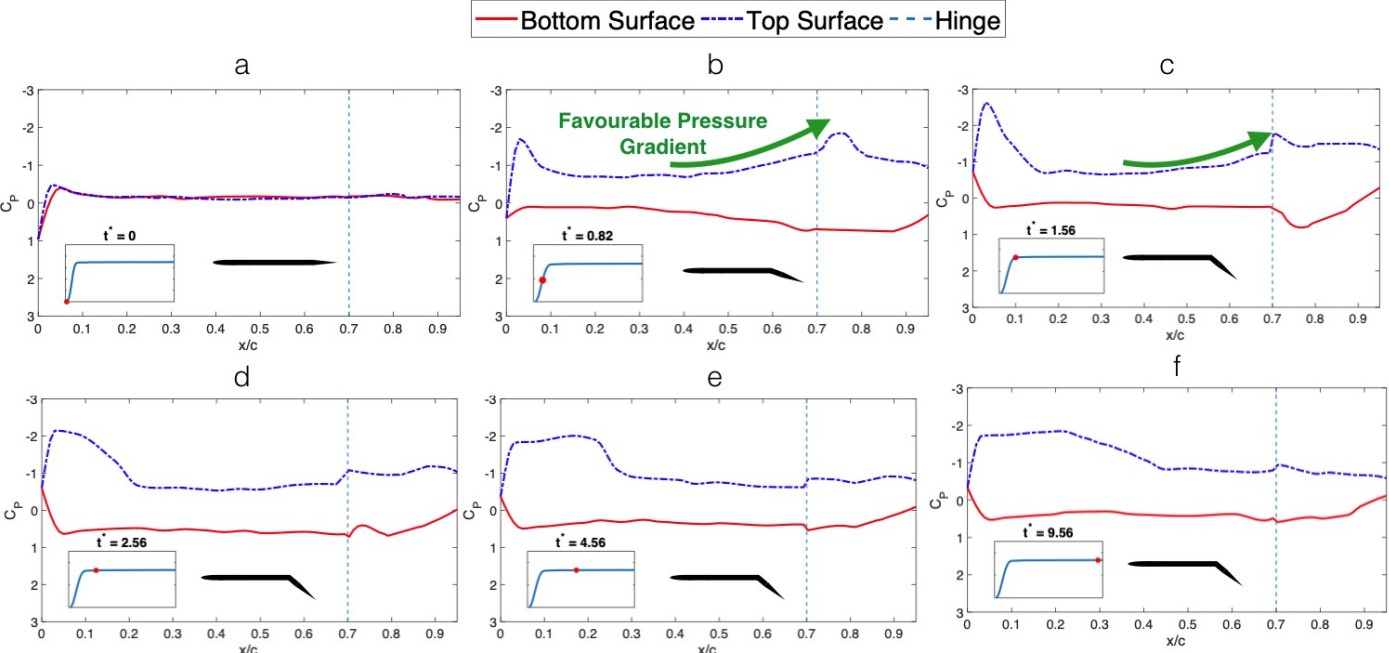

**Figure 7.** $C_P$ distribution during various stages of TECS deflection from $0°$ to $40°$ at $\dot{\beta} = 0.27$.

Figure 8 shows the pressure distribution as the TECS deflection is reversed (i.e., returning from $40°$ to $0°$). The $C_P$ distribution on the top surface shows a region of constant pressure on the front of the airfoil, indicating a region of separation around the leading edge present at the deflected position; see Figure 8a. As the deflection progresses, this region of separated flow reforms to a single suction peak on the leading edge; see Figure 8d. Towards the end of the deflection, the suction peak reduces in magnitude, and the pressure on the top surface and bottom surface again becomes nearly identical; see Figure 8f. The flow on the bottom surface conforms with the changing camber during the deflection and remains attached at the undeflected position. The transition from a separated flow to the attached flow was thus relatively smoother with fewer and smaller vortical structures [23].

Theodorsen's unsteady aerodynamic theory suggests that the transient $C_L$ peak produced within a relatively short convective time during TECS deflection arrives from the deflection rate-dependent lift sources (virtual camber and rotation-induced normal acceleration) and added mass [30]. Thus, the relationship between the magnitude of these $C_L$ peaks to the deflection rate will be investigated in the latter sections.

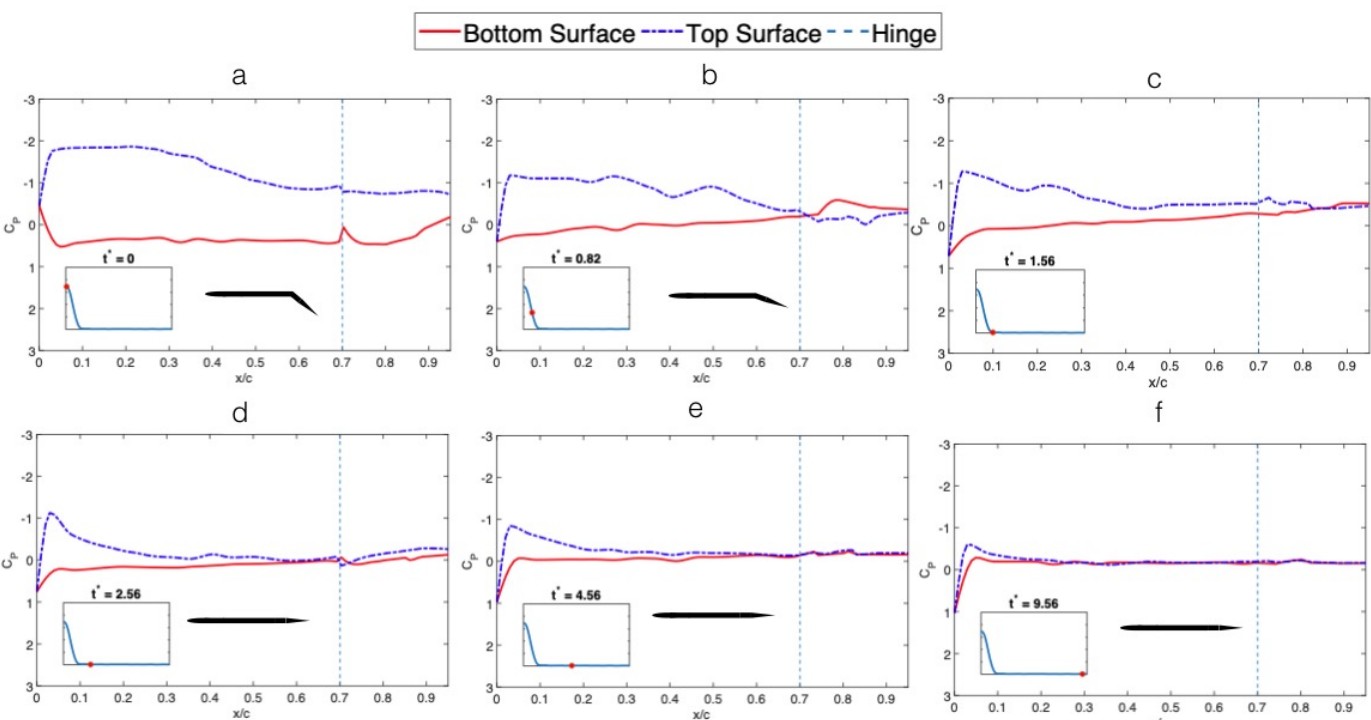

**Figure 8.** $C_P$ distribution during various stages of TECS deflection from $40°$ to $0°$, $\dot{\beta} = 0.27$.

*3.2. Surface Pressure and Flow Behavior during a Dynamic Leading-Edge Control Surface Deflection*

Figure 9 shows the time-varying $C_L$ when deflecting an LECS from $0°$ to $40°$ at $\dot{\psi} = 0.27$. An increase in lift is noticed immediately after the LECS begins to deflect (Figure 9), much like in the case of TECS. Snapshots from smoke flow visualization experiments, presented in Figure 10, show that once the control surface begins to move, the flow conforms to the changing geometry and remains attached. Once the control surface reaches the full deflection, the shear layer rolls up on the upper surface of the airfoil to create an LEV, which grows in size as it convects across the airfoil. This presence of low-core pressure on the top surface can be related to the initial lift peak in Figure 9 (a–c). Concurrently there are also vortical structures that begin to form around the hinge on the bottom surface, which creates a negative lift component. Note that a region of re-circulation was found on the airfoil at higher deflection angles during the time-averaged analysis of statically deflected cases, presented in the static analysis paper [19]. This occurrence of bottom-surface suction can be correlated to the lift being generated in the opposite direction in Figure 9 (d–f).

While the initial LEV on the top surface travels to approximately 30% of the chord, there is additional shedding of smaller vortices from the leading edge. The initial LEV grows in size as it convects towards the trailing edge, forming a larger vortical structure on the upper surface of the airfoil; see Figure 10. In comparison to the vortical structures seen on the bottom surface, the occurrence of vortex mixing can be related to the $C_L$ to increase in the positive direction; see Figure 9 (f–h). This relatively large region of low pressure on the upper surface of the airfoil is only momentary, hence the departure of LEV from the airfoil at $t^* \approx 3$, and the bottom-surface suction contributes to opposing lift generation (dip in the lift for the second time shown in Figure 9 (h–j)). After $t^* \approx 3.5$, the shedding on the LEVs on the upper surface does remain relatively consistent, and $C_L$ gradually decreases and relaxes to a steady-state condition where the flow on the upper surface separates entirely. $C_L$ after $t^* \approx 10$ thus remains settled to a steady-state $C_L$. Towards the end of the transient effect, the smaller LEVs that are shed from the leading edge on the top surface were found to be much further away from the airfoil surface. Though the quantitative contributions to the overall lift from these vortices are not evaluated in this research, it can be said with

confidence that the vortices further away from the airfoil have relatively less impact on the overall $C_L$ than the initial LEV. Hence, the formation and convection characteristics of these vortices play a significant role in lift production over the airfoil.

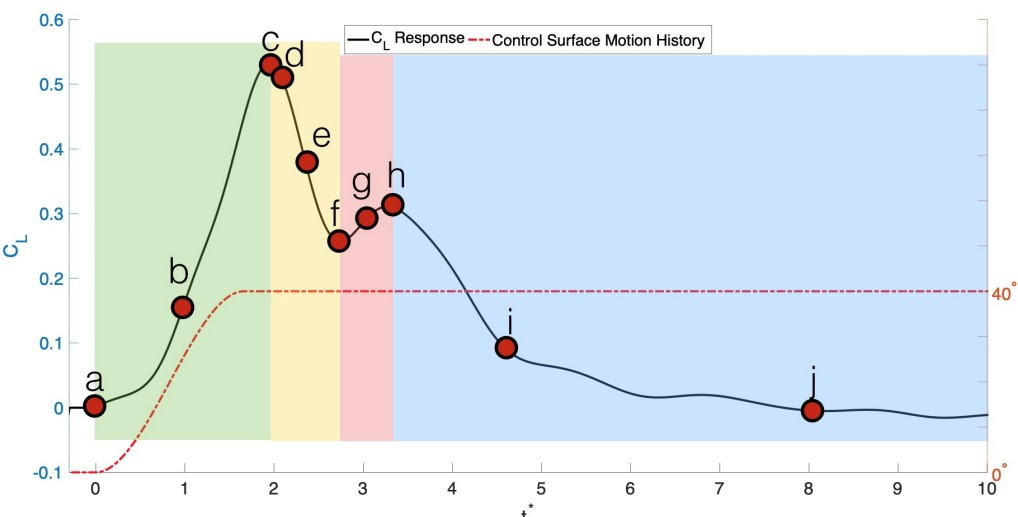

**Figure 9.** Total $C_L$ (i.e., airfoil and control surface) as a function of convective times. Airfoil angle of attack = 0°, LECS deflected from 0° to 40° at a non-dimensional actuation rate, $\dot{\psi} = 0.27$.

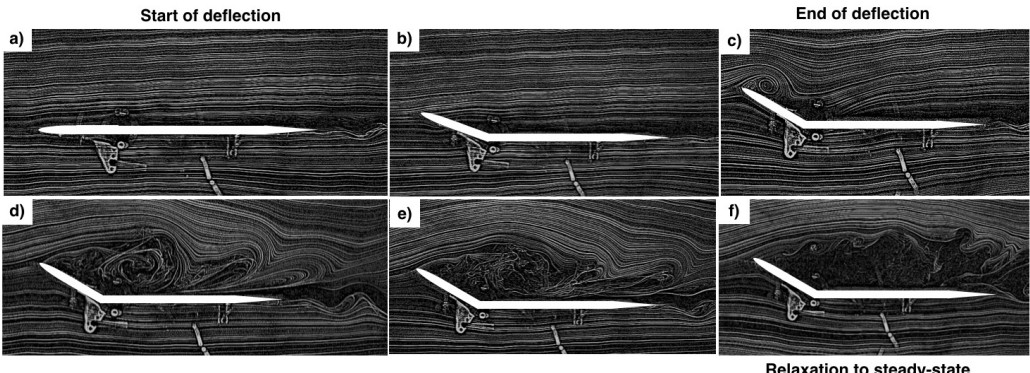

**Figure 10.** Smoke flow visualization of the flow during (**b**) and after the deflection (**c–f**) of a leading-edge control surface. The control horn and the sting seen in the figures above were positioned out of the plane of the smoke flow by 130 mm and did not affect the experiments.

The initial increase in $C_L$, seen in Figure 9, reaches a maximum $C_L$ of 0.55. This occurs at $t* \approx 1.8$. During this stage, the shear layer on the upper surface of the airfoil rolls up to form the initial LEV. As the control surface has just reached full deflection, the region of re-circulation and thus the bottom surface suction does not appear to have occurred. Hence, if the control surface could be returned to the un-deflected position after deflection, this $C_L$ peak could be potentially used without the deleterious consequences (though relatively smaller when compared to the case of TECS) of opposing lift arriving from the loss of initial LEV on the top- and the bottom-surface suction (which forms after the control surface reaches the end of the deflection).

The $C_P$ distribution during LECS deflection at the same rate of $\dot{\psi} = 0.27$ is shown in Figure 11. At an undeflected position, the $C_P$ distributions over the top and bottom surfaces of the airfoil are identical, as expected nominally; see Figure 11a. As the deflection begins, the suction peak on the top surface gradually increases to the end of the deflection. The shear layer beings to roll up and starts forming an LEV, shown by the peak in $-C_P$ on the top surface in Figure 11b. This correlates well with the formation of the LEV seen from Figure 10. After the maximum $C_L$ peak, $C_L$ then reduced to approximately 0.25.

Figure 11d–f shows that this is due to a significant reduction in suction pressure. This is due to the LEVs convecting far away from the top surface of the airfoil. There is a region of re-circulation that starts forming around the hinge on the bottom surface, shown by the peak in $-C_P$ on the bottom surface of the hinge (Figure 11d–f), indicated by small changes to the bottom pressure. However, the largest contribution to lift remains dependent on the upper surface. There is then a small positive trend of $C_L$ seen between $t^* = 2.7$ and 3.2. This is again due to an increase in the suction pressure on the top surface. A suction peak forms at the hinge on the bottom surface, creating a favorable pressure gradient on the bottom surface. Hence, the flow remains attached on the bottom surface throughout most of the transient motion.

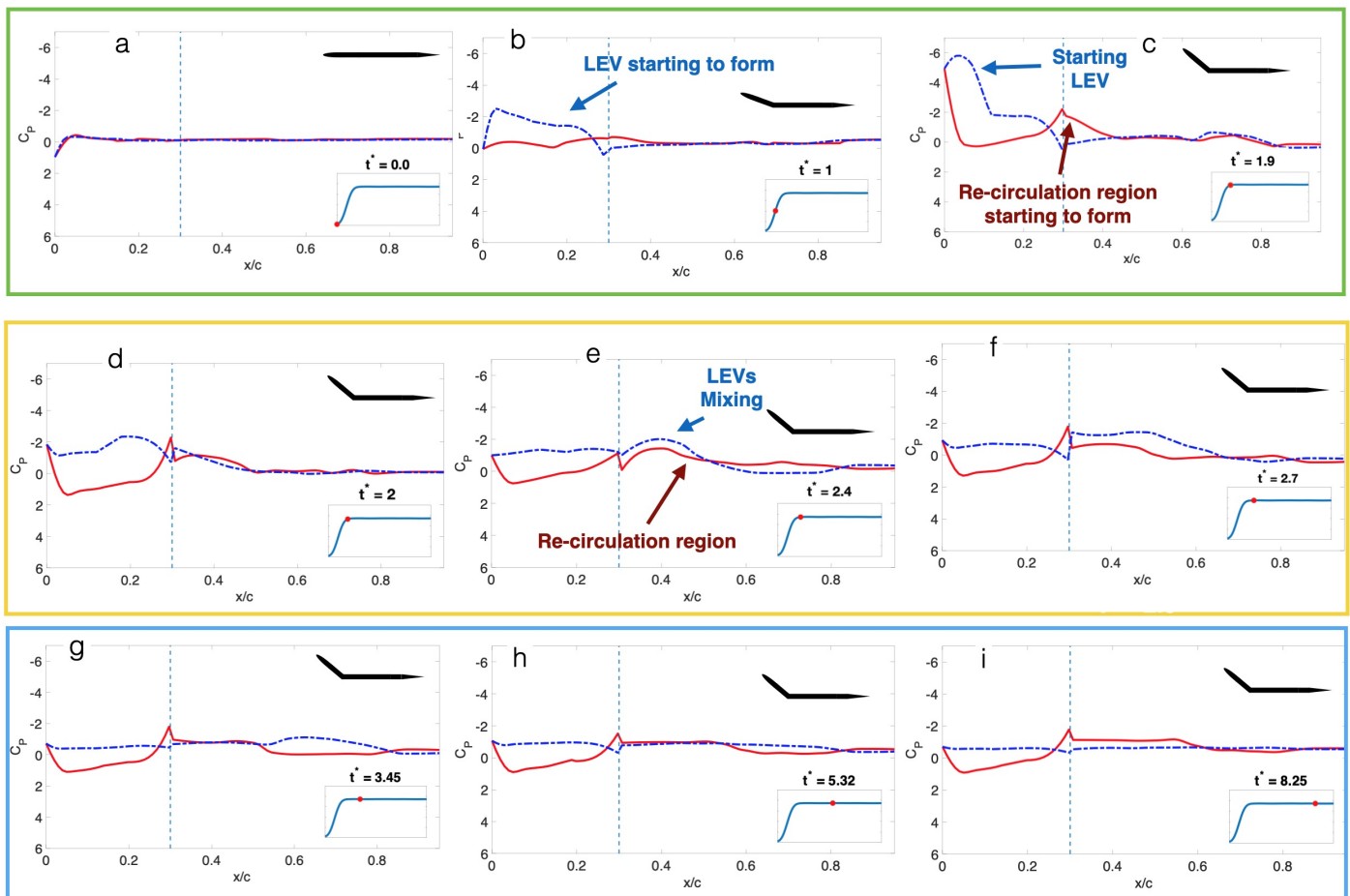

**Figure 11.** $C_P$ distribution during various stages of TECS deflection from $0°$ to $40°$ at $\dot{\psi} = 0.27$.

When the control surface is returned from its deflected position, i.e., $40°$ to $0°$, the flow transitions from being completely separated to attached, and it is observed that the transition consists of formations of relatively smaller vortical structures, much like in the case of TECS. As the deflection begins, the fully separated boundary layer transitions to discrete vortex shedding, after which the flow attaches from the leading edge to the trailing edge. Figure 12 shows surface pressure data during and after the deflection. At $\psi = 40°$, the flow on the top surface is completely separated, shown by relatively flat $C_P$ on Figure 12a. $C_P$ on the bottom surface has a negative suction $C_P$ peak just aft of the leading edge and a suction peak at the hinge, creating a region of favorable pressure gradient on the control surface. Aft of the hinge, the pressure is returning to free-stream pressure. As the deflection initiates, the inflated peak on the bottom surface aft of the leading edge and the suction peak seen at the hinge both gradually reduce in size. The flow settles to a steady condition much quicker (by three convective times) in this case, relative to the previous deflection case from $0°$ to $40°$.

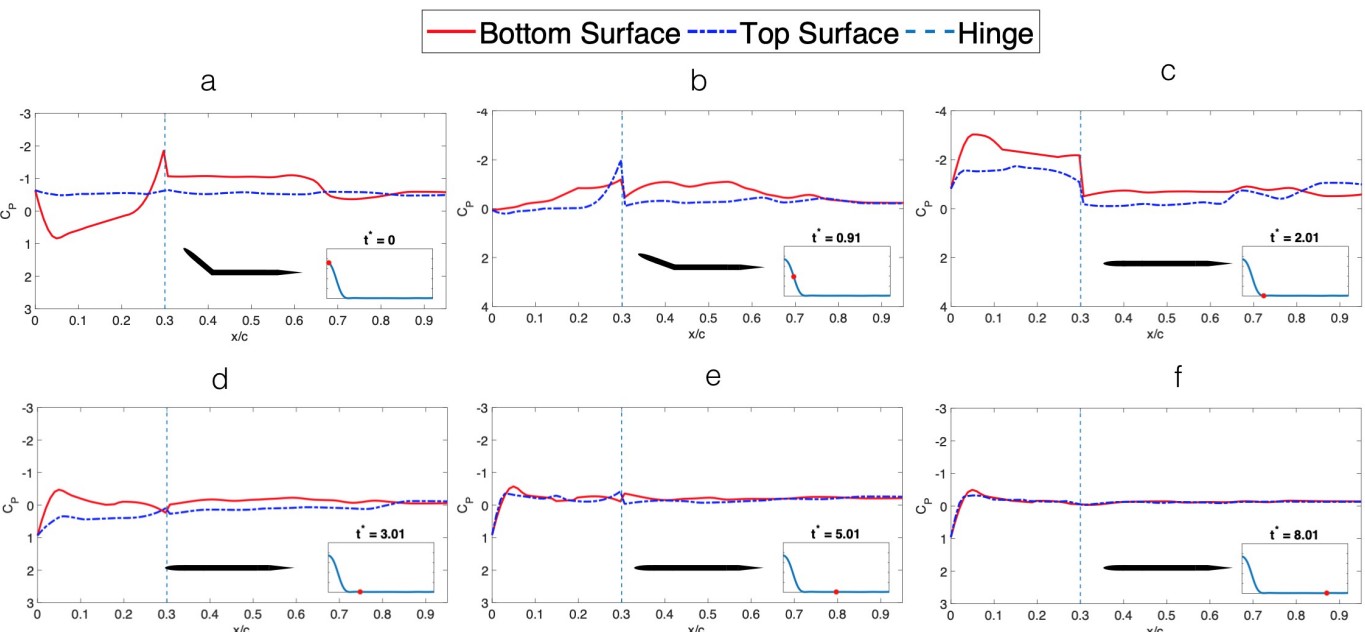

**Figure 12.** $C_P$ distribution during various stages of LECS deflection from 40° to 0° at $\dot{\psi} = 0.27$.

From the investigation of the general flow mechanics associated with dynamic deflections of LECS and TECS, two key conclusions can be drawn. The first is that the lift responds immediately with control surface deflection irrespective of the flow condition, i.e., separated or attached. For both LECS and TECS, it was observed that $C_L$ responded immediately when deflecting from attached to separated flow case and vice versa. The second conclusion is that the formation of vortical structures plays a vital role in the overall duration of the lift response. It was observed that the duration of the unsteady effect (i.e., time for the flow to settle to steady-state) was relatively large for the case of LECS from deflecting from 0° to 40°. This was not the case during deflection from 40° to 0° or in the case of TECS. As the only significant difference between the three cases is the presence of LEVs, the duration of the unsteady effect can thus be linked to the formation of LEVs, which provides a relatively large circulatory lift contribution. Thus, it can be concluded that the $C_L$ response of LECS is a trade-off of positive lift produced on the airfoil by LEVs on the top surface and negative lift generated by the bottom-surface suction.

### 3.3. Effect of Varying Actuation Rates

It is now apparent that part of the transient lift arises from sources that are dependent on the actuation rate of the control surface. The effects of actuation rates on the lift responses of TECS and LECS are studied in this section for the deflecting and returning conditions, i.e., 0° to 40° (attached to separated) and 40° to 0° (separated to attached). The actuation rates selected for this study represent a range of flow conditions around the airfoil, varying from near-steady conditions ($\dot{\beta}, \dot{\psi} = 0.054$) to unsteady flow conditions ($\dot{\beta}, \dot{\psi} = 0.71$).

Figure 13 (left) shows $C_L$ against convective time of dynamic TECS deflections at five different actuation rates. The magnitude of the $C_L$ peak was found to increase significantly with actuation rates. Nearly 1.5 times more lift is produced during this transient period than during similar static deflections, especially at $\dot{\beta} = 0.71$ and 0.54. This is expected to be from deflection rate-dependent lift sources: added mass (arising from initially accelerating the fluid during the motion), virtual camber (variation in normal perturbation velocity), and rotationally induced normal acceleration (from each instantaneous angular deflection). The combination of these actuation rate-dependent lift sources being proportional to acceleration leads to the abrupt rise in the aerodynamic force during the control surface deflection. As expected, during lower actuation rates, the $C_L$ peaks are less obvious and are closer to static values.

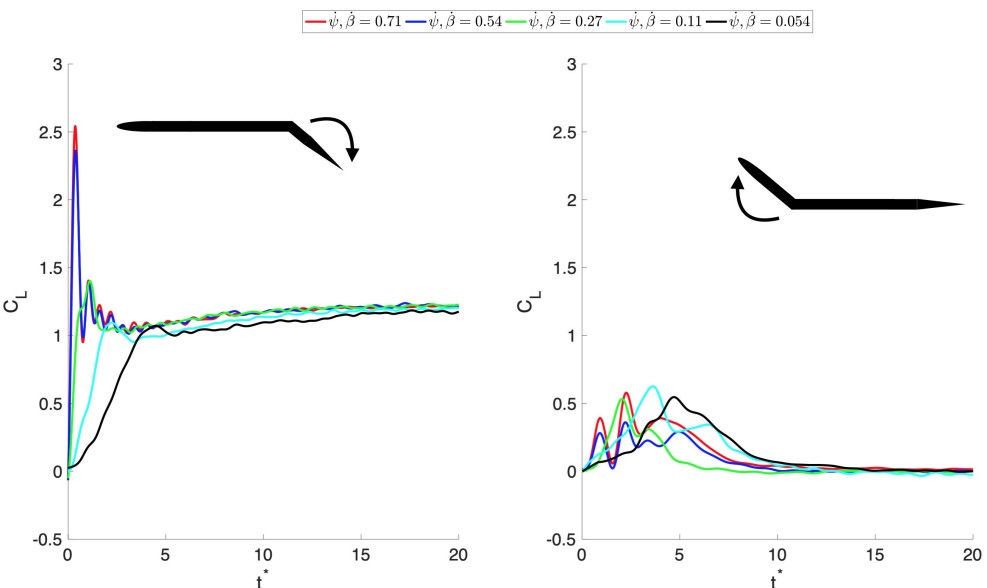

**Figure 13.** Total $C_L$ (i.e., airfoil and control surface) as a function of convective times. Airfoil angle of attack = 0°, control surface TECS (**left**) and LECS (**right**) deflected from 0° to 40° for various deflection rates.

At halfway through the control surface deflection (where the angular velocity is maximum and acceleration is approaching zero), deflection rate-dependent terms are evidently responsible for a large portion of the lift history. The negative trend in control surface acceleration, as it slows down to conclude its motion, likewise correlates to a local lift trough. Around $t^* = 7$, the $C_L$ values from all TECS cases converge to steady-state values. These results correlated well with the findings from a similar analysis of lift response during a rapidly actuated trailing-edge control surface by [31,32].

Figure 14 shows the pressure distribution over the airfoil during various stages of the TECS deflection (top to bottom) and various deflection rates (left to right). For the case of $\dot{\beta} = 0.54$ (left column), it is observed that a favorable pressure gradient exists on the top section during the deflection, which keeps the flow attached to the control surface during the deflection. As the control surface approaches the end of its deflection (subplot d), the area between the top and bottom surface reduces. Post-deflection, the flow on the top surface begins separation (shown by the flat pressure distribution on subplot g) and eventually relates to the steady-state (subplot j). At the higher actuation rate, the area between the top and bottom surface is greater, resolving to a larger transient $C_L$.

Figure 13 (right), presented at the start of this section, also shows five deflection cases at various actuation rates for the LECS configuration. The $C_L$ peaks for all cases vary far less than with TECS. The difference in $C_L$ peak between the fastest rate of $\dot{\psi} = 0.71$ and the slowest rate $\dot{\psi} = 0.054$ is approximately 0.15. However, when compared to the steady values, the $C_L$ peaks are generally around three to four times greater. The PIV vorticity study of dynamically actuated LECS at the same rates (Figure 15) suggests that dynamic actuation of LECS leads to the formation of LEVs on the top surface, which are known to improve the lift on the airfoil while they remain over the airfoil. It is thus evident that, with LECS, even at slower actuation rates, it is possible to achieve relatively large transient control forces. For practical applications, this could mean that once gust is detected, LECS is deflected at a realizable rate and then returned back to the undeflected position at an even more leisurely rate. Unlike the slower actuation rates, multiple peaks are observed during faster actuation ($\geq \dot{\psi} = 0.11$); see Figure 13 (right). As discussed previously, with $\dot{\psi} = 0.27$, the formation and convection of the LEVs are responsible for the fluctuation in the $C_L$ response. With higher actuation rates, large dominant LEVs are formed, which have different convection characteristics (they convect relatively faster) than

smaller LEVs. Hence, the $C_L$ fluctuations for $\dot{\psi} = 0.71$ and $\dot{\psi} = 0.54$ show a relatively significant $C_L$ fluctuation during the transient stage of the $C_L$ response.

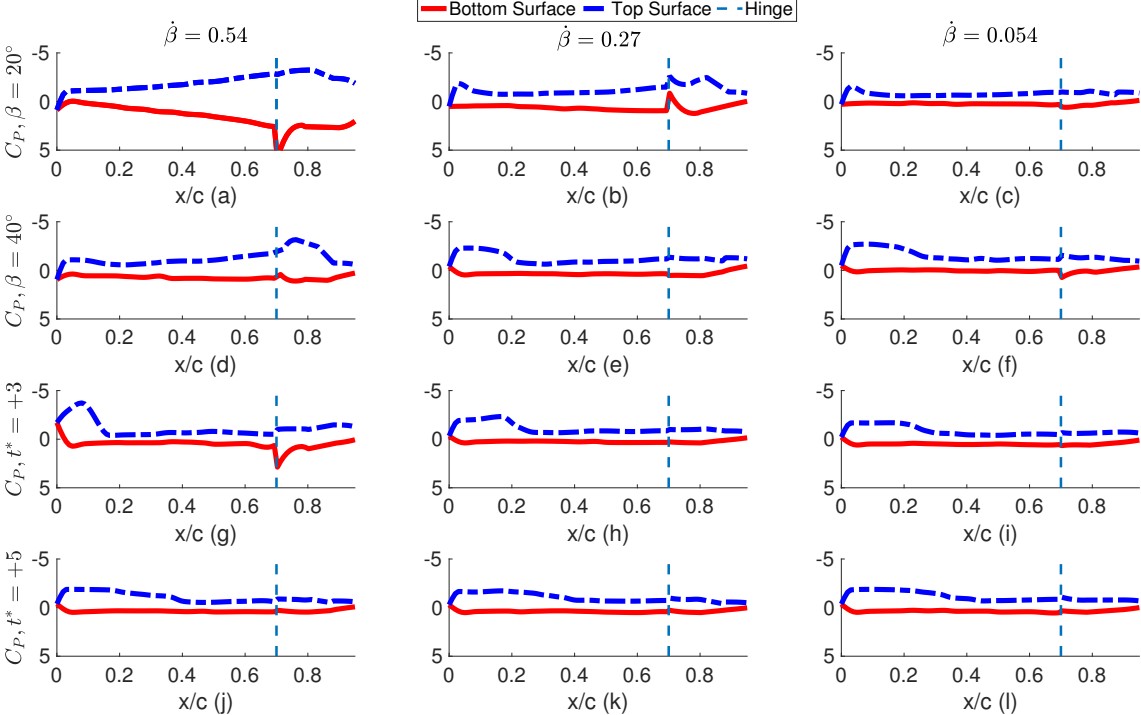

**Figure 14.** $C_P$ distribution during various stages of TECS deflection from $0°$ to $40°$ at various deflection rates. Across the column, pressure distributions are shown at (i) halfway through the deflection, (ii) upon reaching the end of the deflection, (iii) three convective times post-deflection, and (iv) five convective times post-deflection.

Figure 15 displays a vorticity flow field at various stages of the control surface deflection. In all cases, the leading-edge deflection activates the formation of an LEV. LEVs are seen to grow in size as they convect downstream with the flow. It was found that fast actuation rates produced relatively large dominating LEVs, whereas slow actuation rates were found to be largely separated and featured small LEV shedding (Figure 10). During a fast actuation rate, ($\dot{\psi} \geq 0.11$), a classical LEV is formed, dominating the flow, and it is hypothesized that the pitch component of a shear layer aids LEV formation. At slow deflection rates, flow generally separates, and the dominance of LEV is not seen. This presents an interesting question as to whether the development of upper surface flow and LEVs are also dependent on the local angle of attack at the leading edge. The study by [31], where TECS was dynamically actuated from $0°$ to $30°$, addressed this particular question and found the transient lift/pitching moment response to be similar, whether the fore element was at an incidence of $0°$ or $20°$. Investigations of the effect of actuation rates on the LEV size were also previously carried out by [33–38]. In these investigations, it was found that increasing the pitch rate delayed the formation of LEVs on the upper surface and made the LEVs more compact and stronger, thus producing a much larger transient $C_L$.

Experiments were performed in the same manner as actuating oppositely. Figure 16 shows the deflection of TECS (left) and LECS (right) returning from a deflected position, $40°$, to an undeflected position, $0°$. As the deflection rate increases, there is a progression toward a larger amplitude of transient lift increment; this holds true whether the initial flow condition is attached or separated. Recall that the initial condition is not attached flow, yet there is still an instantaneous response. However, the unsteady lift response here differs in magnitude to the transient peak when compared to the lift response going from the attached flow case. Much like the previous case, deflections from separated to attached flow cases are similarly driven by rate-dependent forces, which considerably surpass the

contribution from quasi-steady circulatory lift and moment and result in large lift peaks during the motion transient.

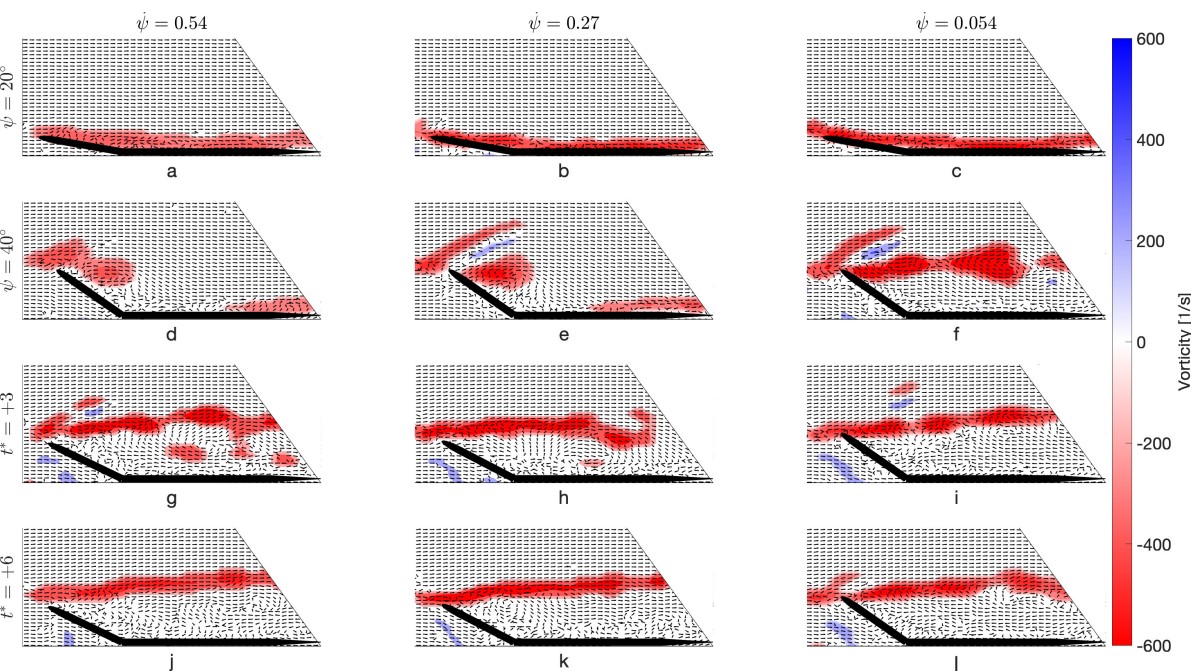

**Figure 15.** PIV vorticity fields of LECS deflection from 0° to 40° at various deflection stages (**top to bottom**) and rates (**left to right**).

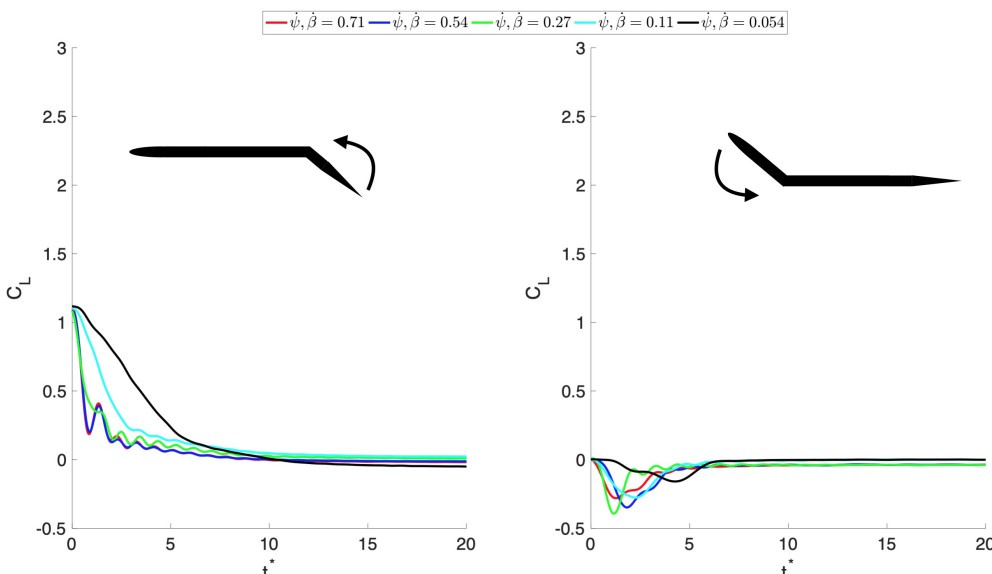

**Figure 16.** Total $C_L$ (i.e., airfoil and control surface) as a function of convective times. Airfoil angle of attack = 0°, control surface TECS (**left**) and LECS (**right**) deflected from 40° to 0° for various deflection rates.

Figure 17 shows the PIV vorticity fields for various LECS deflection cases.

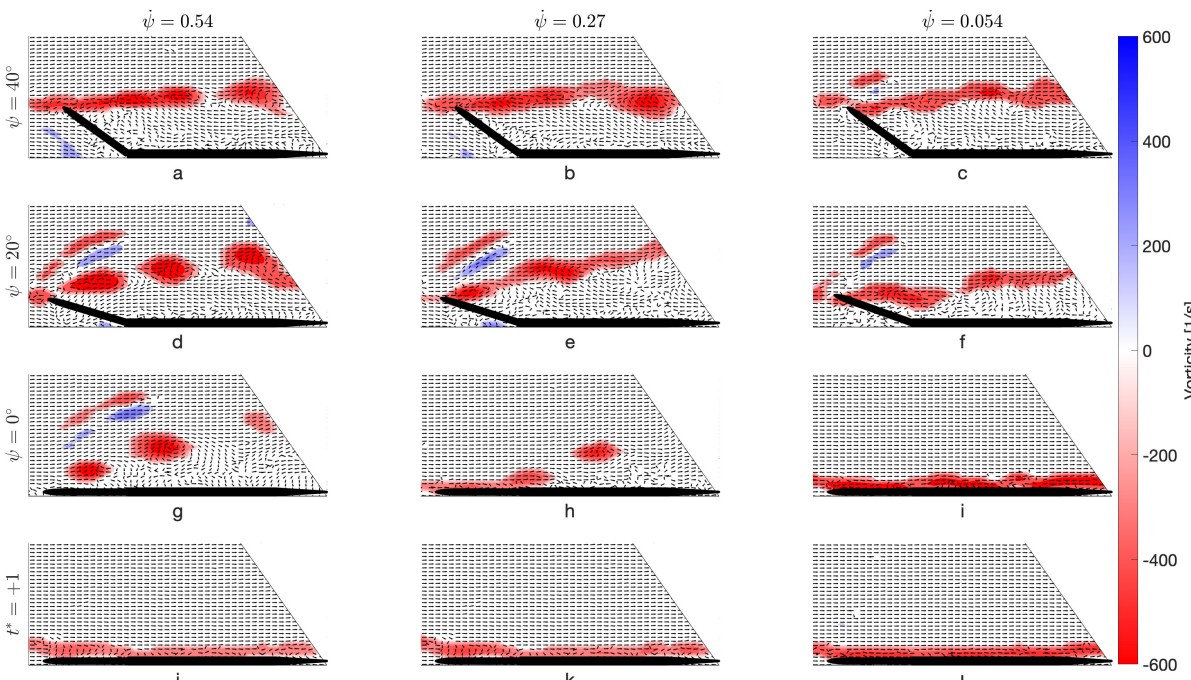

**Figure 17.** PIV vorticity fields of LECS deflection from 40° to 0° at various deflection stages (**top to bottom**) and rates (**left to right**).

Figure 18 summarizes the $C_L$ response for TECS and LECS when deflecting from attached flow case to separated flow case (red) and vice versa (blue). The green line on the figure shows the difference in lift response between steady state and transition from separated to attached flow case. Classical aerodynamic theory using methods of linear superposition would predict a similar result between control surface deflections from 0° to 40° and 40° to 0°; however, we see here that this is not the case. It is hypothesized that the differences (green line in Figure 18) arise from the fact that the case of control surface deflection from 0° to 40° begins its motion from a fully developed steady state, whereas the case of control surface deflection from 40° to 0° experiences a brief period of "inviscid effectiveness," where it reaches a maximum steady circulatory lift contribution. This is evidenced by the large overshoot of the green curve at early times ($t^* \leq 2$). When approaching the steady-state condition, the green line converges to the steady value at around $t^* = 6$ for both TECS and LECS; see Figure 18. Thus, it can be concluded that when the end state is attached flow, relaxation to the steady state is relatively faster.

It is now apparent that the actuation rate of control surfaces or reduced frequency does have a vital influence on the lift response of the airfoil for TECS configuration. The lift response with LECS was found to be almost independent of the motion rate itself. The analyses of the lift responses of LECS and TECS provided in the preceding sections are for a case where the airfoil is aligned to the flow direction, i.e., airfoil angle of attack of zero. This was done so that the focus of the investigation remained close to the immediate unsteady effects arising from varying control surface deflection rates and flow speeds. For TECS, the end condition has zero lift if and only if the control surface comes to rest at zero airfoil angle of attack with respect to the corresponding zero-lift line. When fixed-wing drones are flying in turbulence, the effective angle of attack is constantly changing; thus, TECS must be returned back to a neutral position immediately. On the contrary, LECS produced $C_L$ of nearly zero with the control surface at a positive angle. Thus, LECS could be used more like a fine-tuning device for gust response. However, to validate this hypothesis, further investigations are required at various airfoil angles of attack and deflection angles.

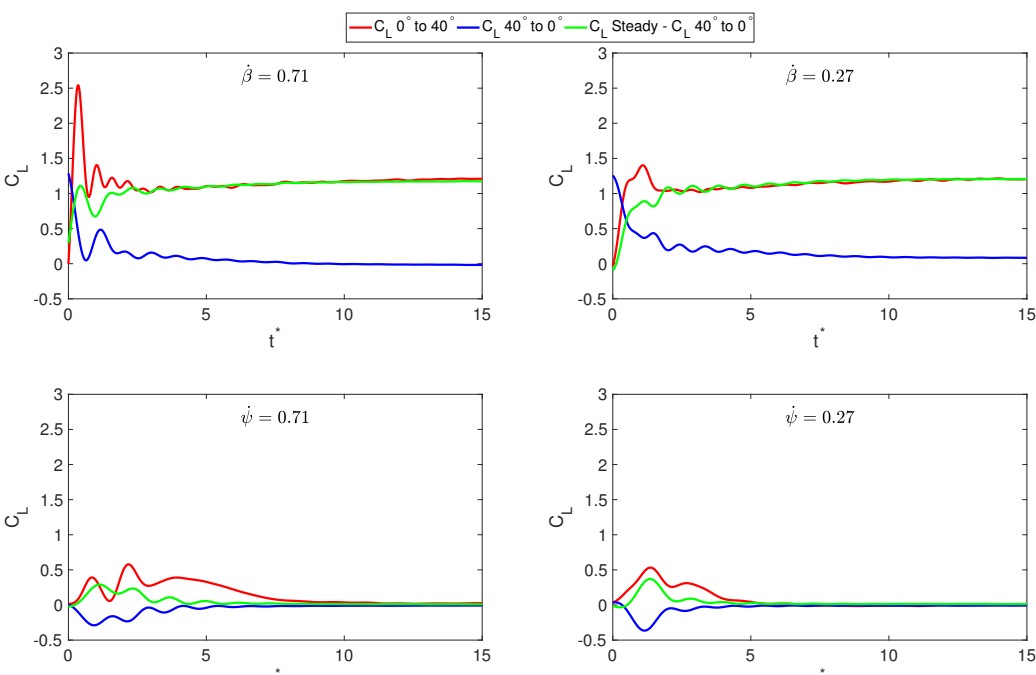

**Figure 18.** Comparison of $C_L$ between deflecting from separated flow to attached flow and from attached flow to separated flow for TECS (**top**) and LECS (**bottom**).

### 3.4. Examination of $C_L$ and $C_M$ on Individual Airfoil Elements

A comprehensive understanding of the fundamental forces acting on the control surface and stationary element of the wing is required to capture these forces theoretically. Thus, in the following section, the $C_L$ distribution over the two segments of the airfoil is investigated at variations in control surface deflection rates at a fixed $\alpha = 0°$.

Figure 19 shows the breakdown of $C_L$ for the fixed and moving elements at four different actuation rates. It is observed that slow to moderate control surface deflections do not have much of an effect on the front element. However, at higher deflection rates, unsteady peaks are also seen on the stationary element. This aligns well with the findings from Mancini's research, which featured a 50% TECS [23]. Figure 20 shows the pitching moment about the quarter chord for TECS (left) and LECS (right) at various deflection rates when deflecting from $0°$ to $40°$. It is observed that (as expected) the trends of $C_M$ are similar to $C_L$ in variations in actuation rates in the case of TECS. This was also the conclusion drawn by a similar previous study [23]; faster actuation rates correlated to larger transient $C_M$. As the $C_P$ in the stationary element is relatively closer to $c/4$ than the $C_P$ of the control surface, it is expected to see a negative $C_M$ for the steady state. During the motion transient, the relative magnitude of $C_L$ between the two airfoil elements changes, with the control surface accounting for almost 50% of $C_L$, which creates negative spikes in $C_M$.

In the case of LECS, the initial lift spike can be attributed almost exclusively to the LECS at sufficiently high actuation rates ($>0.54$); see Figure 21. At lower actuation rates, there exist significant contributions to the transient lift by the stationary rear element as well. Contrary to the case of TECS, the stationary element has a significant contribution to the steady-state lift as well. Thus, the resulting global lift peak can be attributed nearly equally to both the active LECS and the rear stationary element contributions. Furthermore, in the case of LECS, there is no clear relationship between $C_M$ and deflection rates, much like seen in the case of $C_L$. Irrespective of the deflection rate, there is initially a positive $C_M$, which then reverses and eventually relaxes up to a positive $C_M$ just above zero.

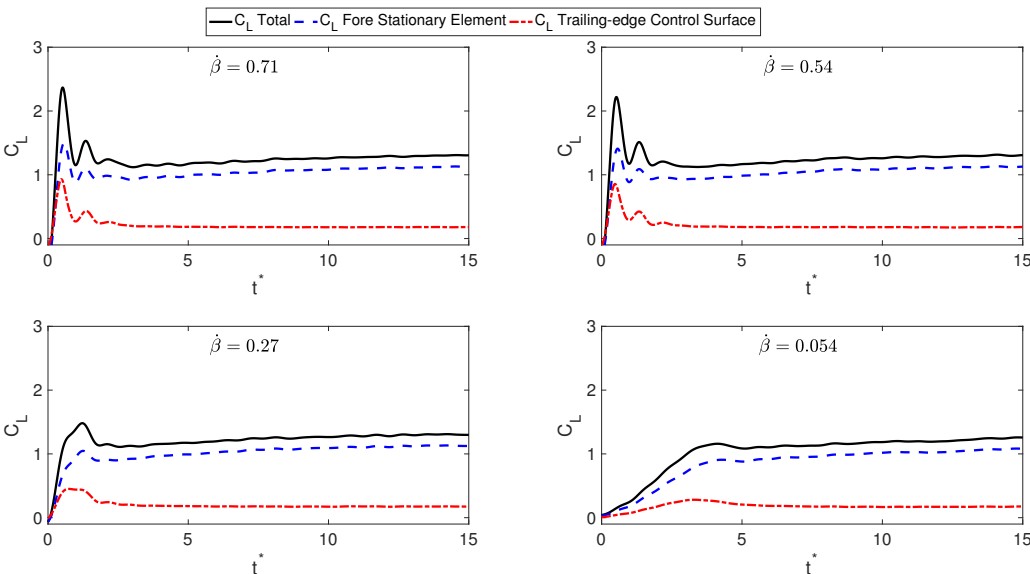

**Figure 19.** $C_L$ produced on TECS and stationary airfoil element at various deflection rates.

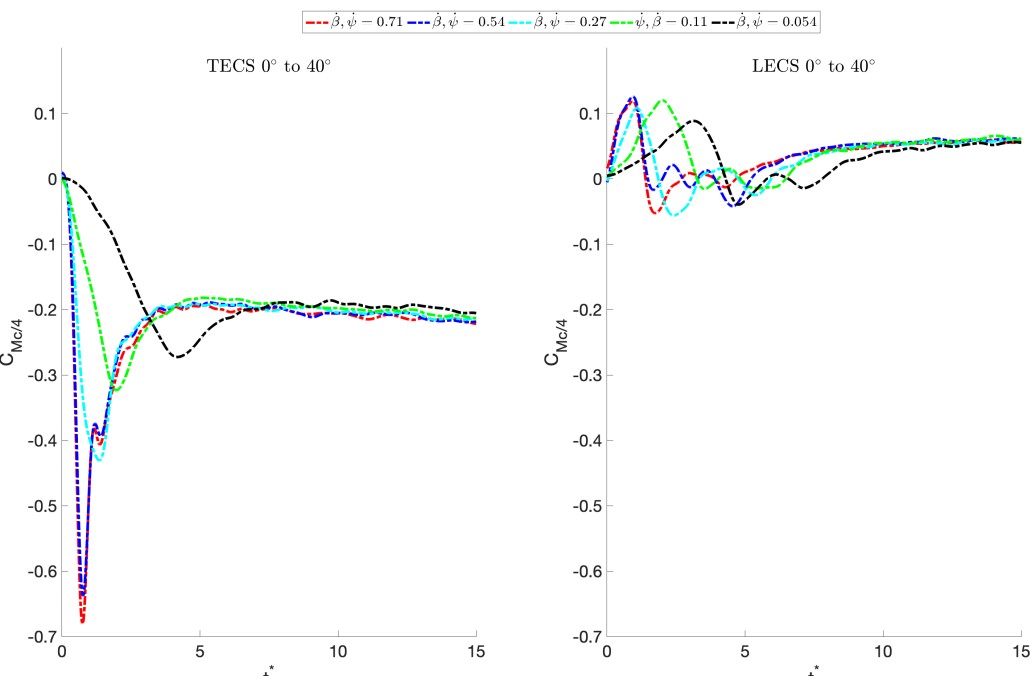

**Figure 20.** $C_M$ for TECS (**left**) and LECS (**right**) from $0°$ to $40°$.

From the preceding sections, a sound understanding of the dynamics of LECS and TECS has been established through experimental investigations. To translate these into control algorithms for small fixed-wing drones with LECS, TECS, or a combination of both, the lift responses of LECS and TECS are compared against existing and modified airfoil theories in order to potentially arrive at simple low-order solutions that can capture the complex dynamics within reasonable accuracy.

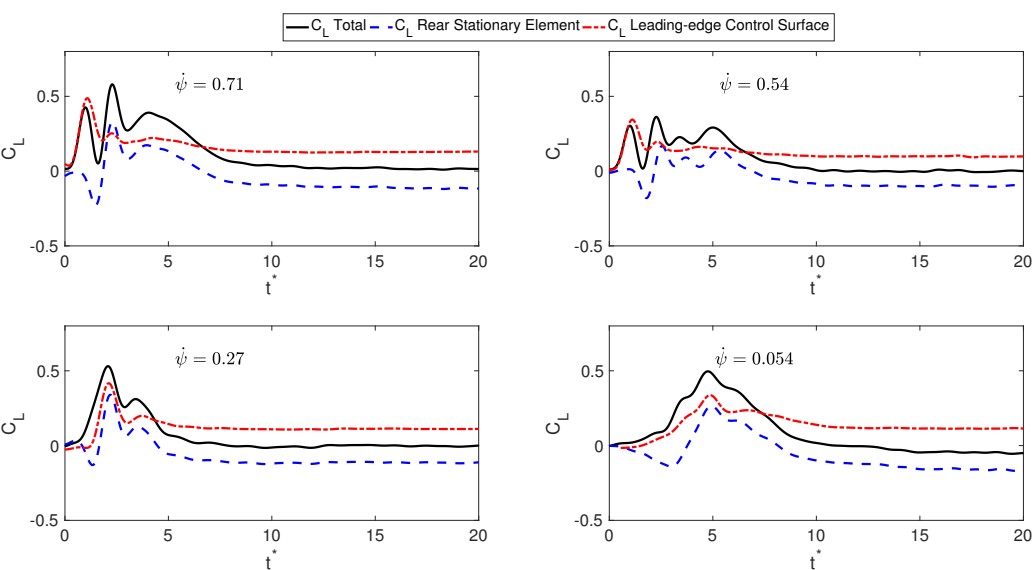

**Figure 21.** $C_L$ produced on LECS and stationary airfoil element at various deflection rates.

*3.5. Analytical Solutions*

The existing aerodynamic solution by Theodorsen is presented in Equation (1) [30], for the case of TECS deflection. The coefficients $T_1, T_4, T_{10}, T_{11}$ depend on airfoil and control surface geometry and $C(k)$ is Theodorsen's function.

$$L_{TECS} = \rho b^2 [u\pi\dot{\alpha} - b\pi a\ddot{\alpha} - UT_4\dot{\beta} - bT_1\ddot{\beta}] + 2\pi\rho ub[u\alpha + b(\frac{1}{2} - a)\dot{\alpha} + \frac{T_{10}}{\pi}u\beta + b\frac{T_{11}}{2\pi}u\dot{\beta}]C(k) \tag{1}$$

Theodorsen's potential flow solution predicts that an airfoil with a deflecting control surface hinged on the trailing edge produces unsteady non-circulatory forces on the front element, despite it remaining stationary throughout the motion. This is due to the distribution of sources and sinks used in the model and placed on the wing to satisfy the no-through-flow boundary condition and thus affects the velocity field everywhere in the flow. Physically, the assumption of incompressibility mandates that any local disturbance to the flow (e.g., physically deflecting a control surface) causes a pressure disturbance everywhere in the flow instantaneously. Theodorsen's model assumes attached, inviscid flow at all times. These two assumptions might be valid during the control surface motion, but not before and after the deflection where the flow is separated.

Figure 22 displays a comparison between the experimentally acquired $C_L$ and Theodorsen's solution. It is evident that Theodorsen's solution (shown by the solid black line) does not accurately capture the experimentally acquired data (solid red line). The magnitude of the $C_L$ peak is underestimated for the case of $\dot{\beta} = 0.71$ and overestimated for $\dot{\beta} = 0.27$ and $\dot{\beta} = 0.054$. Additionally, the comparison reveals a significant overprediction by Theodorsen's solution in the relaxation period of the lift response to the steady state. To gain further insight into this disparity, the individual lift constituents from Theodorsen's solution (shown by dashed lines) are also plotted against convective time in Figure 22. It is evident that the lift force contributions from the steady component are significantly high, as it fails to capture flow separation at the end of the control surface deflection. Likewise, the exponential transition to steady state from Theodorsen's function, $C(k)$, is not accurately captured.

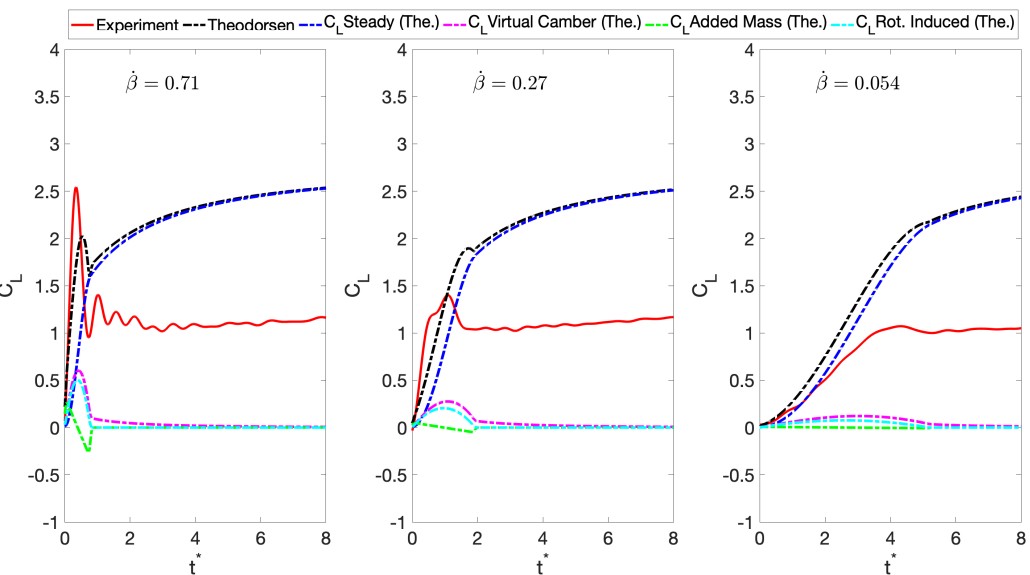

**Figure 22.** Comparison of experimentally acquired $C_L$ against the potential flow solution by Theodorsen [30] at various actuation rates.

Figure 23 shows the comparison between experimentally acquired $C_L$ and the potential flow model derived by Jaworski (Equation (2)). It is observed that the Jaworski model predicts the magnitude and occurrence of the $C_L$ peak reasonably well for the fastest actuation case but fails to do so for the two slower rates. The fluctuation of $C_L$ peaks observed after the initial $C_L$ peak is not captured by the potential flow solution. As discussed earlier, the formation of LEVs on the top surface and formation of the recirculation region on the bottom surface play a significant role in contributing to the unsteady transient lift response of a dynamically actuated LECS. Even well after the control surface comes to the end of the deflection, the LEV formed during the motion convects across the chord of the airfoil, aiding in the generation of lift. Although the unpacking of individual force contributions is not possible with solely surface pressure measurements, it is clear that rate-dependent forces dominate the overall lift production. Hence, this complex balance of vortical structures shed on the top and bottom surface cannot be simply represented with potential flow solutions.

$$L_{LECS} = \rho b^2 [u\pi\dot{\alpha} - b\pi a\ddot{\alpha} - UZ_1\psi + bZ_2\ddot{\psi}] + 2\pi\rho ub[u\alpha + b(\frac{1}{2} - a)\dot{\alpha} + \frac{Z_{13}}{\pi}u\psi + b\frac{Z_{14}}{\pi}u\dot{\psi}]C(k) \qquad (2)$$

A modified approach (Equations (3) and (4)), adopted from [23], was also used to predict the lift response, shown in green in Figure 24. Across the three different actuation rates shown, the lift response prediction is improved. The magnitude of the lift responses and relaxation to the steady state are in good agreement with the experimental data. The modified approach treats the two segments of the airfoil, TECS, and the stationary fore elements separately, assuming that there is no coupling effect between the two. This is not the case in reality, as we have previously observed that deflection of the TECS causes changes in the pressure field on the stationary segment of the airfoil, and there is lift generated by the stationary element. It has been shown that the dynamic deflection of the control surface imposes transient lift generation over the stationary element in the preceding sections. From the experimental results, the magnitude of the lift generated by TECS was found to be relatively lower compared to the lift generated by the stationary element. It is observed there that with Equation (4), which neglects the lift produced by the stationary element, it shows good agreement with experimental results. Coincidentally, ignoring the stationary element and treating TECS essentially as a flat-plate airfoil pivoting about its leading edge made the prediction of the overall lift response "better". This was an interesting finding and was also found by [23]. Hence, before any conclusions could be

drawn regarding validation of this model, the approach would need to be tested across a much larger test matrix featuring a range of control surface sizes.

$$L_{TECS} = -\pi\rho b_{cs}^2(u\dot{\beta} + b_{cs}\ddot{\beta}) + 2\pi\rho u b_{cs}\left(u\beta + \frac{3b_{cs}}{2}\dot{\beta}\right) \tag{3}$$

$$L_{LECS} = -\pi\rho b_{cs}^2(u\dot{\psi} + b_{cs}\ddot{\psi}) + 2\pi\rho u b_{cs}\left(u\psi + \frac{3b_{cs}}{2}\dot{\psi}\right) \tag{4}$$

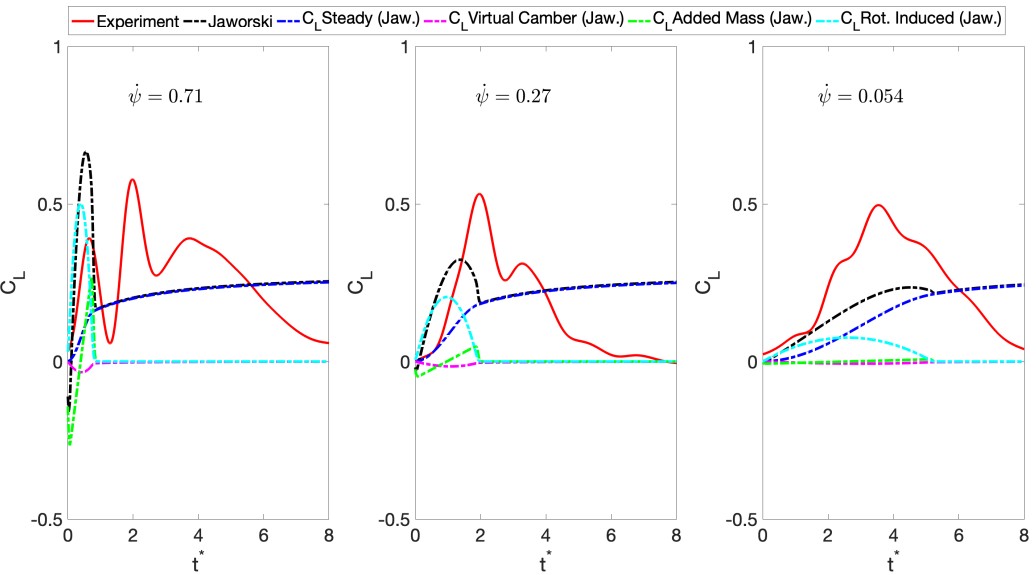

**Figure 23.** Comparison of experimentally acquired $C_L$ against the potential flow model by Jaworski [39] at various actuation rates.

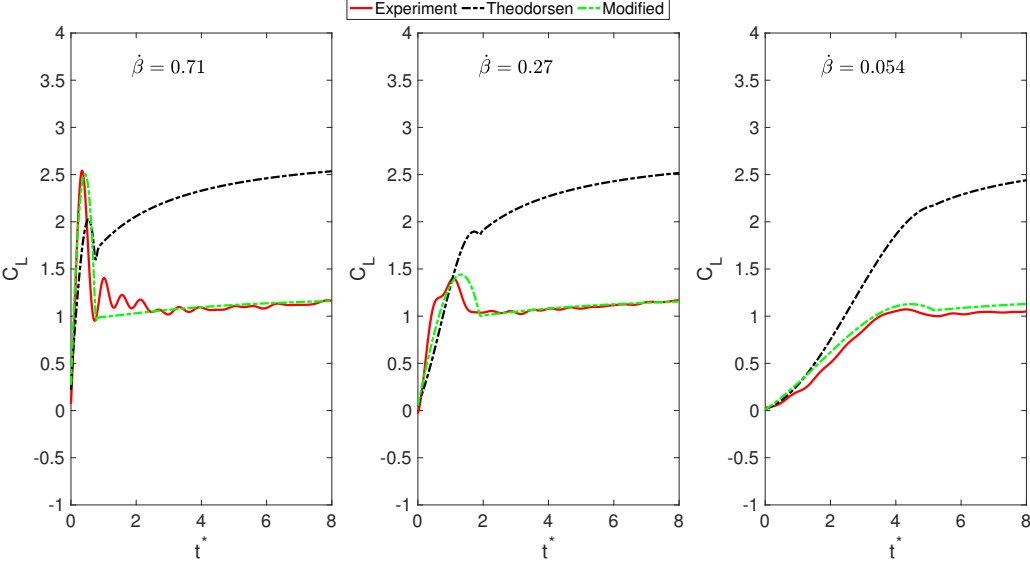

**Figure 24.** Comparison of experimentally acquired $C_L$ against the potential flow model by Theodorsen [30] and a modified model at various actuation rates.

For the modified model, the constants $T_1, T_4, T_{10}$, and $T_{11}$, responsible for relating the geometrical kinematics of the control surface deflections, were removed, as the total lift is now a summation of lift from a rapidly actuating flat-plate airfoil and a stationary element (not producing any lift at the airfoil angle of attack). Thus, the chord of the airfoil is essentially reduced, so the model correlated well with the experimental data.

Figure 25 shows a comparison of $C_L$ from experimental data, Jaworski's solution, and a modified model. It is clear that the modified approach cannot be applied for the case of LECS to predict transient $C_L$. The modified approach, where the control surface and the stationary element are treated as two separate elements, ignoring all coupling effects between them, shows significant disparity with the experimental results. This is unlike the case of TECS, where the modified approach improved the accuracy of estimating the lift response over Theodorsen's solution.

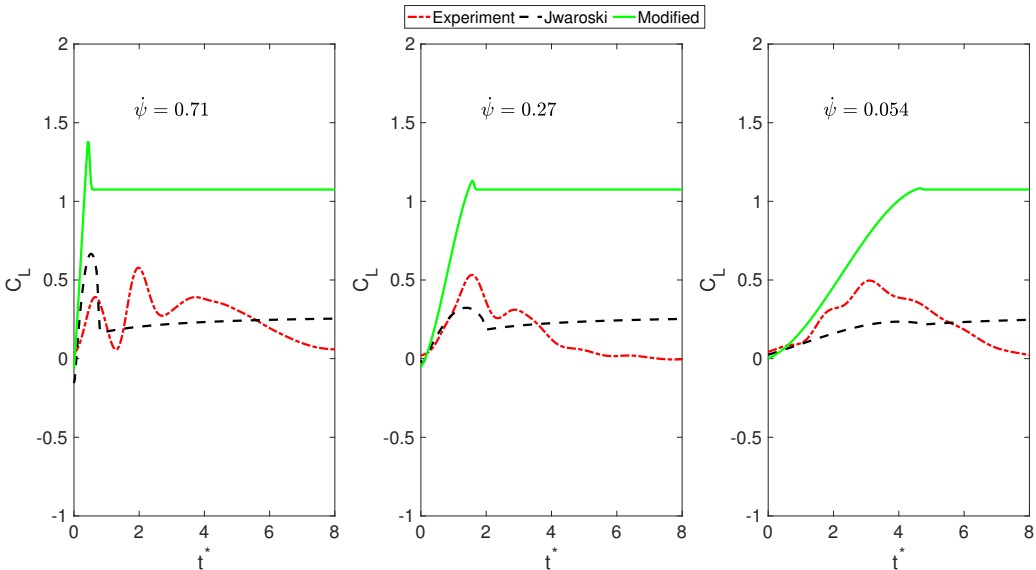

**Figure 25.** Comparison of experimentally acquired $C_L$ against the potential flow model by Jaworski [39] and a modified model at various actuation rates.

As found in the preceding sections, some lift contributions arise from the stationary element. This is because, as the vortex convects across the chord of the airfoil, it is imposing a time-varying low-pressure core, which is generating lift. Hence, there is a lift contribution arising from both the body and the control surface. With the modified approach, there is no lift contribution arising from the stationary element, and all the lift is a function of LECS alone. It overpredicts, because there is nothing to attenuate the lift response. Potential flow does not account for separation, and there is nothing stopping the generation of the steady lift well past the separation point, and the flow relaxes to a steady value.

$C_L$ characteristics of a dynamically deflected LECS were found to be significantly dependent on the convections of vortical structures on the top or bottom surface of the airfoil. These phenomena cannot be captured by potential flow assumptions. The development and shedding of these vortices were found to be dependent on the control surface deflection rate. Thus, rate-dependent lift sources would need to be added into the theoretical models to accurately capture the lift response of dynamically actuated LECS. This can arise from further experiments specifically focused on the dynamic characterization of the development and shedding of vortical structures.

## 4. Concluding Remarks

The study found that leading-edge control surfaces (LECSs) can produce transient $C_L$ nearly three to four times the static values, even with relatively slower actuation rates, while trailing-edge control surfaces (TECSs) had a significant influence on the lift response of the airfoil. However, the maximum transient and steady-state lift produced by TECS was greater than that produced by LECS. Existing theoretical models based on the approach by Theodorsen and a modified model were compared to the experimentally acquired lift responses, and significant differences were found. The modified model was found to improve the accuracy of predicting the lift response of TECS over previous

models. For LECS, the lift response was more affected by the dynamics of the separated region behind the LECS than the inviscid dynamics of the response to the control surface. The study suggests that combined actuation of TECS and LECS could offer better control authority than is currently possible with conventional TECS, especially in gusty flight environments. The authors recommend further investigation into the utilization of changes in wing shape by birds to hold steady flight in gusts.

**Author Contributions:** Methodology, M.M.; validation, A.P. and A.F.; formal analysis, A.P.; Investigation, A.P.; writing—original draft, A.P.; writing—review & editing, A.P. and S.W.; supervision, M.M., A.F., A.M. and S.W.; project administration, S.W. All authors have read and agreed to the published version of the manuscript.

**Funding:** This research was funded by the Australian Post Graduate Award, Australian Federal Government and the Early Research Higher Degree Grant, Defence Science Institute.

**Data Availability Statement:** The datasets generated during and/or analysed during the current study are available from the corresponding author on reasonable request.

**Acknowledgments:** The authors would like to sincerely thank Michael Ol for his advice and guidance on the research project. The authors would like to further extend their gratitude to the Australian Federal Government and the Defence Science Institute for funding this work.

**Conflicts of Interest:** The authors declare that they have no conflicts of interest.

## Nomenclature

| | |
|---|---|
| $\alpha$ | airfoil angle of attack |
| $\beta$ | trailing-edge control surface deflection angle |
| $\dot{\beta}$ | $\frac{\dot{\beta}c}{U}$ non-dimensional trailing edge control surface deflection rate |
| $C_L$ | coefficient of lift |
| $C_D$ | coefficient of drag |
| $C_M$ | coefficient of pitching moment |
| $C_P$ | coefficient of pressure |
| $c$ | chord |
| $U$ | free-stream air flow speed |
| $k$ | $\frac{\omega c}{2U}$ reduced frequency |
| $\psi$ | leading-edge control surface deflection angle |
| $\dot{\psi}$ | $\frac{\dot{\psi}c}{U}$ non-dimensional leading edge control surface deflection rate |
| $Re$ | Reynolds Number |
| $t$ | time |
| $t^*$ | $\frac{tU}{c}$ non-dimensional time (convective time) |

## Appendix A. DPMS Accuracy and Pressure Drift

The DPMS came with the supplier's calibration data; however, this was verified against a calibration test, detailed in Appendix B. The data acquisition time was greater than 10 min per run and as the tunnel had no cooling, drift errors arose from temperature rise. Thus, the effect of drift was corrected for by warming up the DPMS banks and zeroing at the start of each run. An extended test was conducted to demonstrate drift and it was found that the variation in pressure is less than 0.5 Pascals, i.e, $C_P$ of 0.04. Details regarding drift errors have been extensively documented in [40].

## Appendix B. DPMS Calibration

The length of PVC tubing used was long enough to significantly distort pressure fluctuations as they propagated through the tube. This effect is accounted for by utilizing the inverse transfer function, which allows pressure time history at the tap to be accurately reconstructed based on the pressure measured by the time history. This technique has been

utilized by several researchers [24,25] and, when conducted experimentally, also takes into account the dynamic characteristics of the pressure transducers.

To dynamically calibrate the system, a dynamic calibration rig previously designed by [25] was used; see Figure A1. To calibrate an individual pressure tap, the large port (speaker) was placed over the tap on the wing surface, with the O-ring creating a seal. The speaker was then driven with a pre-programmed waveform and data from the transducers sampled at 2 kHz. The fact that the system was sealed allowed the relatively low-frequency sound waves to be transmitted into the tubes with minimal displacement of the speaker cone, thus minimizing distortion. The waveform driving the speaker consisted of a discrete set of 10 frequencies (a 10 Hz fundamental plus nine harmonics) with random phases and had a duration of approximately four seconds. To determine the response function, the signals from the transducer under test and the reference transducer were first truncated to an integer number of periods of the driving waveform. A direct Fourier transform (DFT) was then applied to both signals, and the amplitude and phase of the spectral peaks were compared between the two to determine the transfer function. Further details on the calibration mechanism can be found in [25].

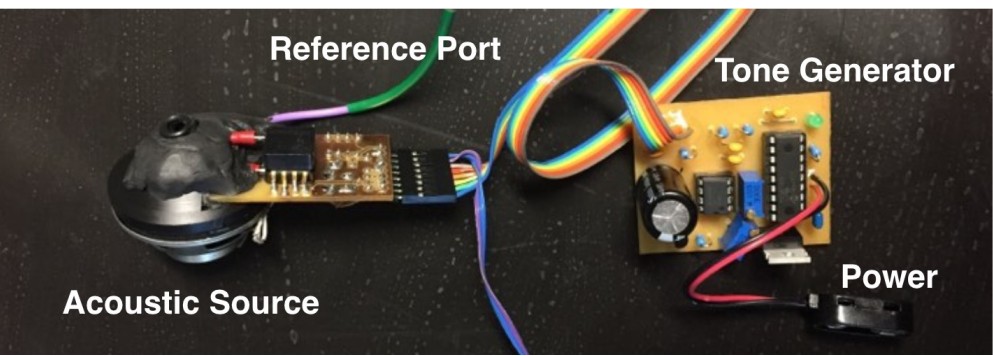

**Figure A1.** Dynamic calibration rig, previously designed by [25].

The taps on each wing were calibrated while connected to the transducer banks in exactly the same manner. The calibration results are presented in Figures A2 and A3. Thus, a single calibration accounted for the combined effects of tap geometry, tubing geometry, transducer dynamic response, transducer static calibration, DAQ card static calibration, and DAQ card inter-channel delay. It also allowed for quick detection of leaking tubes, partially blocked tubes, and malfunctioning transducers. Three calibration runs were performed for each tap, and the average of the three runs was used to obtain a unique transfer function for each tap. Since the calibration only yielded data points at ten discrete frequencies, cubic spline interpolation was used to obtain a continuous transfer function across the whole frequency range. The accuracy of the interpolation was estimated by performing the same procedure on a set of 10 data points obtained analytically and comparing the result to the known analytical transfer function. Across the 0 to 100 Hz range, the maximum error in interpolation was approximately 0.1%, even when inflection/turning points were present in the transfer function.

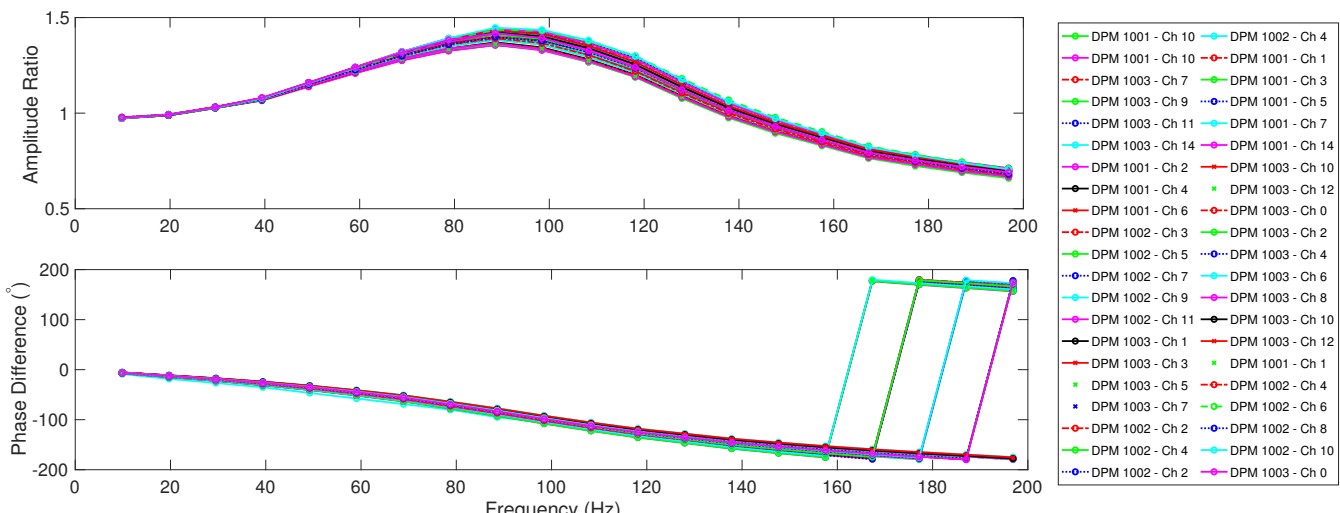

**Figure A2.** Airfoil with TECS: Tube responses of all surface pressure taps.

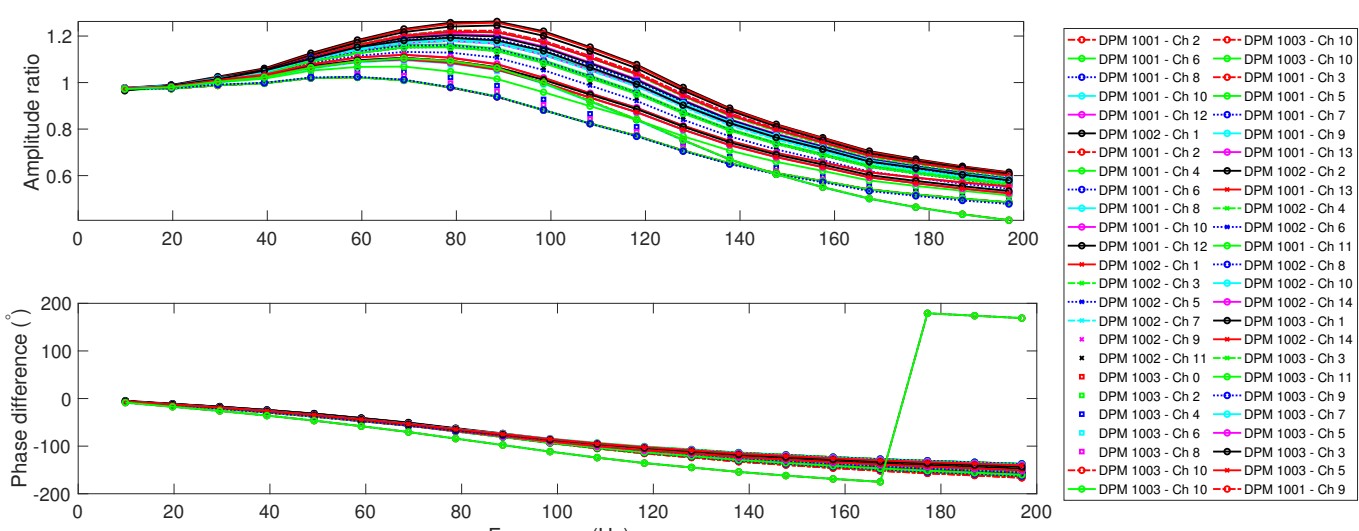

**Figure A3.** Airfoil with LECS: Tube responses of all surface pressure taps.

### Appendix C. Mechanical Vibrations and Noise Interference

A laser Doppler vibrometer (Polytec PSV-400 http://www.vibrometry.co.kr/PSV-40 0_2005.pdf (accessed on 1 June 2018)) was used to analyze the out-of-plane mechanical vibrations associated during the dynamic deflections of the control surfaces. The system can measure vibrations with frequencies up to 1.5 MHz and vibrational velocities of 0.02 μ/s up to 20 m/s. The vibrations were found to be more significant towards the tip of the control surfaces, as they were free to move. The pressure measurements were taken along the center of the span of the airfoil, and the vibrations were found to be less than 0.25° for the fastest deflection case.

### Appendix D. Assessment of Errors in $C_L$ at Fastest Actuation

Figure A4 shows a typical time history of the control surface deflection (right y-axis) and lift responses over an airfoil (left y-axis) at the fastest actuation test case for LECS (black) and TECS (red), starting from $\beta, \psi = 0°$ and ending at 40°. Pressure measurements were sampled at 2 kHz and a low-pass filter was applied in post-processing. Each test was repeated 15 times to document the variation in the data sampled. The variation in $C_L$

was found to be around 2–3% at steady states and approximately 5% at the transient lift peaks. This provides confidence in the data gathered and the processing technique applied. Furthermore, PIV experiments were repeated three times and synchronized to obtain an average frame of vectors for vorticity calculations. The repeatability error was found to be within $\pm 0.15$ m/s and $\pm 30$ 1/s, respectively.

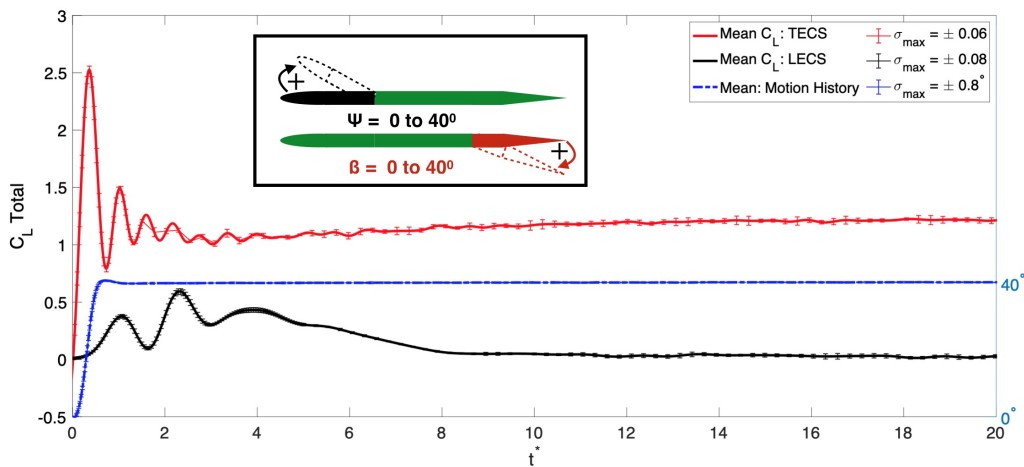

**Figure A4.** Control surface deflection motion history (blue) and lift history of LECS (black) and TECS (red) at $\dot{\beta}, \dot{\psi} = 0.71$.

*Constants*

$$T_1 = -\frac{1}{3}\sqrt{1-b^2}(2+b^2) + b\cos^{-1}b \tag{A1}$$

$$T_4 = -\cos^{-1}b + b\sqrt{1-b^2} \tag{A2}$$

$$T_{10} = \sqrt{1-b^2} + \cos^{-1}b \tag{A3}$$

$$T_{11} = \cos^{-1}b(1-2b) + \sqrt{1-b^2}(2-b) \tag{A4}$$

$$Z_1 = d\sqrt{1-d^2} + (\pi - arcos(d)) \tag{A5}$$

$$Z_2 = \frac{1}{3}(2+d^2)\sqrt{1-d^2} + d(\pi - arcos(d)) \tag{A6}$$

$$Z_{14} = (2d-1)(\pi - arcos(d)) + (2-d)\sqrt{1-d^2} \tag{A7}$$

$$Z_{15} = (2d+1)(\pi - arcos(d)) + (d+2)\sqrt{1-d^2} \tag{A8}$$

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
