# Peer review of "Exploring the Impact of Rapidly Actuated Control Surfaces on Drone Aerodynamics"

_drones, doi:10.3390/drones7080494_

Round 1

Reviewer 1 Report

This paper discusses the  aerodynamic characteristics of rapidly actuated leading-edge and trailing-edge control surfaces for drones. Experiments are setup, and conclusions are presented based on experiment data. However, the paper is not acceptable to publish before certain problems to be clarified. In the following detail my comments:

1. Upon initial review of the paper titled "Improving Control and Manoeuvrability of Fixed-Wing Drones through High-Speed Control Movements", I anticipated a comprehensive discussion on flight dynamics. However, after a thorough reading, it became evident that the primary focus of the paper was on the study of high-lift devices. Consequently, I believe the current title may not accurately reflect the core content of the paper, leading to a potential disconnect between reader expectations and the actual subject matter.

2. In this paper, the Reynold number is set as Re =40000. How was this number chosen?

3. I believe that there was a missing quote mark for "stable" in Line 40.

4. The normalized actuation rates are set between 0.1 and 1.4 (Line 132). How were these numbers chosen?

5. The paper sets out to examine the aerodynamic characteristics of high-lift devices in the context of fixed-wing drones. However, the size of drones can vary greatly, ranging from a few centimeters to several meters. It remains unclear as to what specific drone size the research in this paper is targeting.

Author Response

Reviewer 1 Comments:

This paper discusses the aerodynamic characteristics of rapidly actuated leading-edge and trailing-edge control surfaces for drones. Experiments are setup, and conclusions are presented based on experiment data. However, the paper is not acceptable to publish before certain problems to be clarified. In the following detail my comments:

  1. Upon initial review of the paper titled "Improving Control and Manoeuvrability of Fixed-Wing Drones through High-Speed Control Movements", I anticipated a comprehensive discussion on flight dynamics. However, after a thorough reading, it became evident that the primary focus of the paper was on the study of high-lift devices. Consequently, I believe the current title may not accurately reflect the core content of the paper, leading to a potential disconnect between reader expectations and the actual subject matter.
  2. In this paper, the Reynold number is set as Re =40000. How was this number chosen?
  3. I believe that there was a missing quote mark for "stable" in Line 40.
  4. The normalized actuation rates are set between 0.1 and 1.4 (Line 132). How were these numbers chosen?
  5. The paper sets out to examine the aerodynamic characteristics of high-lift devices in the context of fixed-wing drones. However, the size of drones can vary greatly, ranging from a few centimeters to several meters. It remains unclear as to what specific drone size the research in this paper is targeting.

Response:

The authors would like to acknowledge the reviewer for their valuable comments, which helped us to improve the quality of the manuscript. The following changes have been made:

  1. The authors have suggested a revised title. However, this paper aims to explore the fluid mechanics of leading and trailing edge control surfaces primarily used for generating lift for control, rather than functioning as high lift devices. The authors have conducted extensive literature review of leading-edge flaps/ control surfaces in the context of high lift devices and flow control techniques. For interested readers, please see Panta A, Mohamed A, Marino M, Watkins S, Fisher A. Unconventional control solutions for small, fixed wing unmanned aircraft. Progress in Aerospace Sciences. 2018 Oct 1;102:122-35
  2. As mentioned in line 34, “A chord-based Reynolds number of 40,000 was selected”. This represents the typical flight speed of small, fixed-wing UAS. The scope of the paper was for low Reynolds. For the interested reader, on the effect of various Reynolds Numbers, please see Panta, A., Watkins, S., Marino, M., Fisher, A. and Mohamed, A., 2019. Lift response of rapidly actuated leading-edge and trailing-edge control surfaces for MAVs. In AIAA SciTech 2019 Forum (p. 1397).
  3. The missing quote mark has been added to the manuscript, in Line 40.
  4. A variety of rates were chosen to examine different aerodynamic effects, ranging from fully steady to unsteady flow behavior. The study revealed that actuation rates exceeding 0.1 initiates the formation of unsteady flow structures. Consequently, a range of these rates was selected to facilitate the generation and evolution of prominent flow structures.
  5. As mentioned in Line 1, this study specifically concentrates on small fixed-wing unmanned aircraft systems (UAS). The experiments were carried out at a chord-based Reynolds number of 40,000, corresponding to a chord length of 15 cm. To simulate 2D flow conditions, the airfoils used in the experiments were equipped with end plates. Therefore, the findings presented in the paper are not restricted to a particular wingspan.

Reviewer 2 Report

This work presents experimental results and discussion regarding rapidly actuated leading-edge and trailing-edge control surfaces. The differences obtained from classical and modified unsteady aerodynamic models point out that better models are required to improve the control authority of small fixed-wing drones. Comments and suggestions to improve the paper are given below.   Contents   In Section 2, please add an explanation or figure depicting how the chordwise pressure transducers were installed in the model.   Lines 194-209 please correlate the explanations in this paragraph with letters (a) through (f) in Figure 6, similarly to what was done with Figure 7.   The paragraph between lines 392-402 needs some clarification. The authors mention Figure 15 and 16, however I believe the discussion is only about the LECS (Figure 16), since it says "response here is similar in shape and differs in magnitude", but the response of the TECS has very different shapes comparing deflection increasing and decreasing. I got the impression that Figure 15 was not analyzed in the paper. Also, I don't quite agree with "large lift spikes during the motion transient" (line 402) in none of the figures. Perhaps the authors could clarify that part as well.   The explanation on Line 405 needs correction: "The green line on the figure shows the difference between the two deflections". It shows the difference between steady-state and the blue line. (or, it is the negative value of the blue line, plus the steady-state value).   The conclusion on line  415-417 is misleading: "Thus it can be concluded that when the end state is attached flow, relaxation to the steady-state is much faster." For TECS, Figure 17 shows me that both curves (red and green) settle down at similar times, whereas for LECS, I agree that the green line settles down faster than the red line. It is not clear why the green line is necessary in this figure. All the conclusions using the green line could be drawn from the blue line.   In Lines 424-425: "For TECS, the end condition has zero lift if and only if the control surface comes to rest at zero airfoil angle of attack." I did not understand what you meant by "rest at zero airfoil angle of attack". With a deflected flap, the lift will be zero if the angle of attack is zero with respect to the corresponding zero-lift line. Is that what you meant?   In Line 447, "As the lift is greater on the stationary element, it is expected to see a negative CM peak throughout the deflection history." I don't quite agree with the explanation. I believe it is more related to the moment arm and the lift distribution around the quarter-chord. The center of pressure (CP) in the stationary element is probably much closer to "c/4" than the CP of the control surface. That's why the steady-state value of Cm is negative. During transients, the relative magnitude of CL between the two parts changes, with the control surface accounting for almost 50% of CL, which creates negative spikes in Cm.           Format/English   Line 81: "RAWT is a closed-return tunnel and has a hexagonal test section of 2.1m, 1.3m, and" hexagonal or octagonal?   Line 169 "Due to the high number of repeats, any bias uncertainties get averaged out and imposes a negligible influence on the results" impose   Line 172 "thus gives confidence in the results presented" give   Line 274 "At an undeflected position, the CP distribution over the top and bottom surface of the airfoil is identical" are identical   Line 392 "Figure 15 and 16 shows the deflection" show   Line 463: "complex dynamics within reasonable accurately."  reasonable accuracy?   Line 535: "Figure 24 shows of comparison of CL from experimental..." should be "[...] shows a comparison..."

Author Response

Reviewer 2 Comments:

This work presents experimental results and discussion regarding rapidly actuated leading-edge and trailing-edge control surfaces. The differences obtained from classical and modified unsteady aerodynamic models point out that better models are required to improve the control authority of small fixed-wing drones. Comments and suggestions to improve the paper are given below.  

  1. Contents   In Section 2, please add an explanation or figure depicting how the chordwise pressure transducers were installed in the model.  
  2. Lines 194-209 please correlate the explanations in this paragraph with letters (a) through (f) in Figure 6, similarly to what was done with Figure 7.  
  3. The paragraph between lines 392-402 needs some clarification.
  4. The authors mention Figure 15 and 16, however I believe the discussion is only about the LECS (Figure 16), since it says "response here is similar in shape and differs in magnitude", but the response of the TECS has very different shapes comparing deflection increasing and decreasing. I got the impression that Figure 15 was not analyzed in the paper. Also, I don't quite agree with "large lift spikes during the motion transient" (line 402) in none of the figures. Perhaps the authors could clarify that part as well.  
  5. The explanation on Line 405 needs correction: "The green line on the figure shows the difference between the two deflections". It shows the difference between steady-state and the blue line. (or, it is the negative value of the blue line, plus the steady-state value).  
  6. The conclusion on line  415-417 is misleading: "Thus it can be concluded that when the end state is attached flow, relaxation to the steady-state is much faster." For TECS, Figure 17 shows me that both curves (red and green) settle down at similar times, whereas for LECS, I agree that the green line settles down faster than the red line. It is not clear why the green line is necessary in this figure. All the conclusions using the green line could be drawn from the blue line.  
  7. In Lines 424-425: "For TECS, the end condition has zero lift if and only if the control surface comes to rest at zero airfoil angle of attack." I did not understand what you meant by "rest at zero airfoil angle of attack". With a deflected flap, the lift will be zero if the angle of attack is zero with respect to the corresponding zero-lift line. Is that what you meant?  
  8. In Line 447, "As the lift is greater on the stationary element, it is expected to see a negative CM peak throughout the deflection history." I don't quite agree with the explanation. I believe it is more related to the moment arm and the lift distribution around the quarter-chord. The center of pressure (CP) in the stationary element is probably much closer to "c/4" than the CP of the control surface. That's why the steady-state value of Cm is negative. During transients, the relative magnitude of CL between the two parts changes, with the control surface accounting for almost 50% of CL, which creates negative spikes in Cm.          
  9. Line 81: "RAWT is a closed-return tunnel and has a hexagonal test section of 2.1m, 1.3m, and" hexagonal or octagonal?  
  10. Line 169 "Due to the high number of repeats, any bias uncertainties get averaged out and imposes a negligible influence on the results" impose  
  11. Line 172 "thus gives confidence in the results presented" give  
  12. Line 274 "At an undeflected position, the CP distribution over the top and bottom surface of the airfoil is identical" are identical  
  13. Line 392 "Figure 15 and 16 shows the deflection" show  
  14. Line 463: "complex dynamics within reasonable accurately."  reasonable accuracy?  
  15. Line 535: "Figure 24 shows of comparison of CL from experimental..." should be "[...] shows a comparison..."

Response:

The authors would like to acknowledge the reviewer for their valuable comments, which helped us to improve the quality of the manuscript. The following changes have been made:

  1. In the revised manuscript, Figure 1 has been included, accompanied by additional text in lines 75-85, to provide a more detailed elaboration on the airfoil setup.
  2. In the revised manuscript, additional text has been incorporated to establish correlations between the explanations in the paragraph and the subfigure labels (a) through (f) shown in lines 204 to 220. This helps to enhance the clarity and understanding of the figures.
  3. To enhance clarity, additional text and figure sub-labels have been included in lines 204 to 220 of the revised manuscript.
  4. The reference to the figures has been amended for clarification, see Line 405. Further amendments have been made to the text to improve the clarity of the section. 
  5. Agree. The text in line 418 in the revised manuscript (previously 405) has been amended and presented in red font.
  6. Agree. The content in lines 427-429 (previously lines 415-417) has been revised. While it is acknowledged that many conclusions from the green line can be inferred from the blue line, the authors are aiming to identify and quantify the fundamental sources of lift to ultimately develop a low-order numerical model that can arcuately characterize the transient lift characteristics.
  7. Agree. The text in lines 436-438 (previously Lines 424-425) has been amended for clarification.  Yes, the authors indented to mean that “the lift will be zero if the angle of attack is zero with respect to the corresponding zero-lift line”.
  8. The authors are appreciative of the valuable feedback and agree with it. As a result, additional text has been inserted in lines 459 to 462 to address the feedback in a more comprehensive manner.
  9. This grammatical error has been amended.
  10. This grammatical error has been amended.
  11. This grammatical error has been amended.
  12. This grammatical error has been amended.
  13. This grammatical error has been amended.
  14. This grammatical error has been amended.
  15. This grammatical error has been amended.

Reviewer 3 Report

This study proposes a method of the use of rapidly actuated leading-edge and trailing-edge control surfaces to improve the control authority of small fixed-wing drones. Although the obtained results are intriguing, some revisions are necessary before considering this paper for publication. The following suggestions are provided:

1. The main contributions of this paper should be clearly presented in the Introduction section.

2. The article lacks a clear demonstration of the main process involved in the proposed method. It is important to establish a clear connection between the innovative points presented in the paper.

3. In Fig.4, the explanation of selecting these parameters should be given, e.g., dot{beta} and dot{psi},

4. In Fig.24, the theoretical results seem to be very different from the experimental results, please give the corresponding reason.

5. Some references are relatively out of date, papers in recent three years should be cited.

Extensive editing of English language required.

Author Response

Reviewer 3 Comments:

This study proposes a method of the use of rapidly actuated leading-edge and trailing-edge control surfaces to improve the control authority of small fixed-wing drones. Although the obtained results are intriguing, some revisions are necessary before considering this paper for publication. The following suggestions are provided:

  1. The main contributions of this paper should be clearly presented in the Introduction section.
  2. The article lacks a clear demonstration of the main process involved in the proposed method. It is important to establish a clear connection between the innovative points presented in the paper.
  3. In Fig.4, the explanation of selecting these parameters should be given, e.g., dot{beta} and dot{psi},
  4. In Fig.24, the theoretical results seem to be very different from the experimental results, please give the corresponding reason.
  5. Some references are relatively out of date, papers in recent three years should be cited.
  6. Extensive editing of English language required.

Response:

The authors would like to acknowledge the reviewer for their valuable comments, which helped us to improve the quality of the manuscript. The following changes have been made:

  1. This has been outlined in line 70 – 74, in the Introduction Section.
  2. Section 2.2 of the paper provides a detailed explanation of the test methodology, while Section 2.3 covers the specifics of the results processing.
  3. The definitions of parameters are provided in the Nomenclature section. *denotation is used to represent parameters that are non-dimensionalized with respect to chord (such as convective time). dot is used to represent parameters that are non-dimensionalized with respect to speed (i.e..., to get a rate). A variety of rates were chosen to examine different aerodynamic effects, ranging from fully steady to unsteady flow behavior. The study revealed that actuation rates exceeding 0.1 initiates the formation of unsteady flow structures. Consequently, a range of these rates was selected to facilitate the generation and evolution of prominent flow structures.
  1. The rationale for the disparity is provided in Lines 552 to 553. For additional reading on the fundamental of the unsteady forces, interested readers can find further information in Holger Babinsky, Robbie J Stevens, Anya R Jones, Luis P Bernal, and Michael V Ol. Low order modelling of lift forces for unsteady pitching and surging wings. In 54th AIAA Aerospace Sciences Meeting, page 0290, 2016.
  2. The manuscript includes relevant references found from a comprehensive literature survey, the most recent ones dating from 2019.
  3. The manuscript has undergone necessary revisions to enhance grammar and overall clarity.

Reviewer 4 Report

A discrepancy exists between the title and the content as detailed in the abstract. The title implies a focus on enhancing the control and maneuverability of fixed-wing drones through high-speed control movements. However, the abstract primarily explores the impact of rapidly actuated control surfaces on drone aerodynamics. If the manuscript extensively discusses the drones' control and maneuverability, consider revising the abstract to incorporate these aspects for consistency. Conversely, if the main focus lies on the control surfaces and their aerodynamic effects, the title might require a revision to accurately portray the manuscript's primary content.

Please ensure that Figure 1 is inserted into the text after its mention, not before.

In line 124, the reference to the motor link appears to be extraneous. Please consider its removal.

Similarly, please review line 143 for redundant content.

In line 149, the necessity for a 1.5-hour warm-up period for the sensors is asserted. Please provide evidence or relevant references supporting this statement.

It is commonly understood that direct force measurements often yield greater accuracy than those derived from pressure distribution. Could you clarify the reasoning behind the chosen methodology of pressure measurements?

To enhance understanding, please consider including an image of the wing showing the holes. Additionally, provide specifications such as airfoil thickness, the diameter of the pressure tubes, and the method of their installation on the upper and lower surfaces.

Figure 5 seems to lack explanation for points e and f. Moreover, the cause for the continuous lift increase after point f, including its potential duration, requires clarification. The results post point c appear counterintuitive and could potentially be an artefact of filter application on the pressure sensor signals.

Similarly, in Figure 8, the lift consistency between points a and j raises questions. If the lift doesn't change after 5 seconds, what advantage does the leading edge control surface (LECS) offer for control?

The difference between Figure 12 and Figure 14 is not clear. Please clarify.

The paper's structure and flow seem challenging to follow, impacting the readability. Consider reviewing and revising for smoother transitions and logical organization.

The inclusion of section 3.5 "Analytical Solutions" seems out of place. Could you explain its relevance to the overall paper?

Figure 21 is rather complex and difficult to interpret. Simplifying the figure or providing a more detailed explanation could help improve readers' understanding.

Lastly, it appears that both the figures and their respective explanations need further refinement to enhance reader comprehension. Please pay careful attention to improving these aspects.

I hope these comments are helpful for refining your manuscript.

Please check the typo in the text. For example in the title: 'Manoeuvrability'.

Author Response

Reviewer 4 Comments:

  1. A discrepancy exists between the title and the content as detailed in the abstract. The title implies a focus on enhancing the control and maneuverability of fixed-wing drones through high-speed control movements. However, the abstract primarily explores the impact of rapidly actuated control surfaces on drone aerodynamics. If the manuscript extensively discusses the drones' control and maneuverability, consider revising the abstract to incorporate these aspects for consistency. Conversely, if the main focus lies on the control surfaces and their aerodynamic effects, the title might require a revision to accurately portray the manuscript's primary content.
  2. Please ensure that Figure 1 is inserted into the text after its mention, not before. 
  3. In line 124, the reference to the motor link appears to be extraneous. Please consider its removal. 
  4. Similarly, please review line 143 for redundant content.
  5. In line 149, the necessity for a 1.5-hour warm-up period for the sensors is asserted. Please provide evidence or relevant references supporting this statement.
  6. It is commonly understood that direct force measurements often yield greater accuracy than those derived from pressure distribution. Could you clarify the reasoning behind the chosen methodology of pressure measurements?
  7. To enhance understanding, please consider including an image of the wing showing the holes. Additionally, provide specifications such as airfoil thickness, the diameter of the pressure tubes, and the method of their installation on the upper and lower surfaces.
  8. Figure 5 seems to lack explanation for points e and f. Moreover, the cause for the continuous lift increase after point f, including its potential duration, requires clarification. The results post point c appear counterintuitive and could potentially be an artefact of filter application on the pressure sensor signals.
  9. Similarly, in Figure 8, the lift consistency between points a and j raises questions. If the lift doesn't change after 5 seconds, what advantage does the leading edge control surface (LECS) offer for control?
  10. The difference between Figure 12 and Figure 14 is not clear. Please clarify.
  11. The paper's structure and flow seem challenging to follow, impacting the readability. Consider reviewing and revising for smoother transitions and logical organization.
  12. The inclusion of section 3.5 "Analytical Solutions" seems out of place. Could you explain its relevance to the overall paper?
  13. Figure 21 is rather complex and difficult to interpret. Simplifying the figure or providing a more detailed explanation could help improve readers' understanding.
  14. Lastly, it appears that both the figures and their respective explanations need further refinement to enhance reader comprehension. Please pay careful attention to improving these aspects. I hope these comments are helpful for refining your manuscript.

Comments on the Quality of English Language

  1. Please check the typo in the text. For example in the title: 'Manoeuvrability'.

Response:

The authors would like to acknowledge the reviewer for their valuable comments, which helped us to improve the quality of the manuscript. The following changes have been made:

  1. Agree, the authors have proposed a revised title.
  2. The positioning of the figures and text is controlled by the Journal’s template to improve the content density of the page.
  3. Agree, the hyperlink to the motor and the potentiometer has been removed.
  4. Agree, the hyperlink to the motor and the potentiometer has been removed.
  5. To investigate drift errors, an extended test was performed. Pressure recordings were collected at various time intervals from the start-up phase. It was observed that a warm-up time of 90 minutes effectively eliminated drift. The test results can be made available as supplemental data upon request.
  6. Pressure measurement provides several measurements over direct force measurements, including:
    1. Detection of localized flow variations and pressure gradients makes it particularly useful for studying complex flow phenomena like separation, and vortex shedding.
    2. Provides detailed spatial resolution and offers insights into the local aerodynamic forces acting on different sections of the wing.
    3. Non-intrusive measurement: Pressure sensors can be attached to the surface of an object without significantly altering its shape or aerodynamic characteristics. This non-intrusive nature minimizes interference with the flow field, ensuring that the tested object's integrity is maintained during experimentation.
  7. In the revised manuscript, Figure 1 has been included, accompanied by additional text in lines 75-85, to provide a more detailed elaboration on the airfoil setup.
  8. Lines 191-200 (in the revised manuscript) have been revised to enhance clarity and readability. Notably, during the dynamic deflection, Trailing-Edge Vortices (TEVs) is shed. Upon reaching the end of the deflection, a large TEV is initially formed, followed by the shedding of several smaller TEVs. The x-axis in Figure 6 represents convective time.
  9. The pressure measurements presented in the manuscript were obtained by averaging data from multiple test runs, taking into account any potential errors. Further details can be found in Appendix D. It is important to note that the significance of rapidly actuated TECS lies in the response and magnitude of transient lift, which directly contributes to the rapid maneuverability and control.
  10. We are uncertain about the meaning of a reviewer's comment, it would be helpful to provide the specific comment or context as Figures 12(right) and 14 (Figures 13 and 15 in the revised manuscript) are both displaying results for TECS deflection case from 0 to 40 degrees, at varying actuation rates.  
  11. We have made a fairly extensive revision to the manuscript to ensure improved readability and a smoother flow throughout.
  12. The development of a low-order numerical model will facilitate the implementation of flight control algorithms, ensuring effective control of small fixed-wing unmanned aircraft systems (UAS) with leading and trailing-edge control surfaces. In Section 3.5, the manuscript provides valuable insights by comparing experimental results with numerical models that accurately represent the unsteady lift response resulting from rapid actuation of individual control surfaces or their combined operation. This is outlined in Lines 472 – 477 in the revised manuscript.
  13. Figure 22 in the revised manuscript (previously Figure 21) details the breakdown of lift constituents as shown in the Figure legend. Significant revisions have been made to lines 493-502 to enhance clarity and readability.
  14. Significant revisions have been made to lines 493-502 to enhance clarity and readability.
  15. The grammatical error has been amended.

Round 2

Reviewer 3 Report

No more comments.